# On the Impacts of the Random Initialization in the Neural Tangent Kernel Theory

**Guhan Chen**
Department of Statistics and Data Science
Tsinghua University
Beijing, China
chen-gh23@mails.tsinghua.edu.cn

**Yicheng Li**
Department of Statistics and Data Science
Tsinghua University
Beijing, China
liyc22@mails.tsinghua.edu.cn

**Qian Lin**[*]
Department of Statistics and Data Science
Tsinghua University
Beijing, China
qianlin@tsinghua.edu.cn

## Abstract

This paper aims to discuss the impact of random initialization of neural networks in the neural tangent kernel (NTK) theory, which is ignored by most recent works in the NTK theory. It is well known that as the network's width tends to infinity, the neural network with random initialization converges to a Gaussian process $f^{\mathrm{GP}}$, which takes values in $L^2(\mathcal{X})$, where $\mathcal{X}$ is the domain of the data. In contrast, to adopt the traditional theory of kernel regression, most recent works introduced a special mirrored architecture and a mirrored (random) initialization to ensure the network's output is identically zero at initialization. Therefore, it remains a question whether the conventional setting and mirrored initialization would make wide neural networks exhibit different generalization capabilities. In this paper, we first show that the training dynamics of the gradient flow of neural networks with random initialization converge uniformly to that of the corresponding NTK regression with random initialization $f^{\mathrm{GP}}$. We then show that $\mathbf{P}(f^{\mathrm{GP}} \in [\mathcal{H}^{\mathrm{NT}}]^s) = 1$ for any $s < \frac{3}{d+1}$ and $\mathbf{P}(f^{\mathrm{GP}} \in [\mathcal{H}^{\mathrm{NT}}]^s) = 0$ for any $s \geq \frac{3}{d+1}$, where $[\mathcal{H}^{\mathrm{NT}}]^s$ is the real interpolation space of the RKHS $\mathcal{H}^{\mathrm{NT}}$ associated with the NTK. Consequently, the generalization error of the wide neural network trained by gradient descent is $\Omega(n^{-\frac{3}{d+3}})$, and it still suffers from the curse of dimensionality. On one hand, the result highlights the benefits of mirror initialization. On the other hand, it implies that NTK theory may not fully explain the superior performance of neural networks.

## 1 Introduction

In recent years, the advancement of neural networks has revolutionized various domains, including computer vision, generative modeling, and others. Notably, large language models like the renowned GPT series [8, 51] have shown exceptional proficiency in language-related tasks. Similarly, neural networks have achieved significant successes in image classification, as evidenced by works such as [27, 34, 37]. This proliferation of neural networks spans a wide range of fields. Despite these

---

[*]Corresponding author

38th Conference on Neural Information Processing Systems (NeurIPS 2024).

impressive achievements, a comprehensive theoretical understanding of why neural networks perform so well remains elusive in the academic community.

Several studies have delved into the theoretical properties of neural networks. Initially, researchers were keen on exploring the expressive capacity of networks, as demonstrated in seminal works like [17, 28]. These studies established the Universal Approximation Theorem, asserting that sufficiently wide networks can approximate any continuous function. More recent research, such as [15, 26, 43] extended this exploration to the effects of deeper and wider network architectures. However, a significant challenge remains in these studies: they often do not fully explain the generalization power of neural networks, which is crucial for evaluating the performance of a statistical model.

Recently, some researchers have examined the generalization properties of networks. Bauer and Kohler [5], Schmidt-Hieber [46] showed the minimax optimality of networks with various activation functions for specific subclasses of Hölder functions, within the nonparametric regression framework. In contrasts to the static ERM approach, some studies made more attention to the dynamics of neural networks, particularly those trained using gradient descent (GD) and stochastic gradient descent (SGD)[2, 13, 20].

With similar insights, Jacot et al. [31] explicitly introduced the Neural Tangent Kernel (NTK) concept, demonstrating that there exists a time-varying neural network kernel (NNK) which converges to a fixed deterministic kernel and remains almost invariant during training as network width approaches infinity. And thus NTK theory proposes that network training can be approximated by a kernel regression problem [4, 29, 39, 50]. As a general case, fully-connected networks directly trained by GD, Lai et al. [35], Li et al. [41] showed the generalization ability of two-layer and multi-layer networks, respectively.

This paper mainly follows [4, 35, 41], and explores the impact of initialization in the NTK theory. Prior research [35, 41] which verified the minimax optimality of network utilized the so-called *mirrored initialization* setting. It refers to a combination of mirrored structure and mirrored initial value of parameters, which results in a zero initial output function. However, the assumption divates from the commonly used initialization strategy in real-world applications, whose initial output is actually non-zero. To bridging the gap, in this study we explore the generalization ability of standard non-zero initialized network, within the NTK theory framework. Our findings reveal that the vanilla non-zero initialization will theoretically results in poor generalization ability of network, especially when the data has relatively large dimension. If that is true, it suggests a divergence between theoretical models and real-world applications, highlighting a potential limitation in the current understanding of the NTK theory. Therefore, we arrive at a critical problem central to this study:

*Does initialization significantly impact the generalization ability of networks within the kernel regime?*

## 1.1 Our contribution

• *Network converges to a NTK predictor uniformly*. We show that under standard initialization, the network function converges to the NTK predictor uniformly over the entire training process and over all possible input in the domain. The convergence is essential in the study of the generalization ability of network in NTK theory. However, in previous work, the initial values of network has long been overlooked. Under mirrored initialization which leads to zero initial output function, Arora et al. [4], Lai et al. [35], Li et al. [41] demonstrated the point-wise convergence and the uniform convergence of network, respectively. More recently, Xu and Zhu [54] studied the uniform convergence of NTK under standard initialization, but did not study the convergence of the network function. Why the initial output of the network is ignored is not that it is insignificant, but rather because it is a stochastic function, making it challenging to analyze in convergence. Our findings make it valid to approximate the network's generalization ability based on the corresponding NTK predictor's performance.

• *The generalization ability of standardly non-zero initialized fully-connected network*. Our research explores the impact of standard non-zero initialization in NTK theory. At this issue, Zhang et al. [56] proposes the existence of implicit bias induced by non-zero initialization, when the neural network is completely overfitted. We delve deeper into this argument, studies the exact formula of the bias at any stage of training, within the framework of NTK theory. Additionally, we established that the (optimally tuned) learning rate of network is $n^{-\frac{3}{d+3}}$, even when the regression function is

sufficiently smooth. This insightful discovery implies a notable limitation in the generalization ability of networks with non-zero initialization, if NTK theory can precisely approximate the performance of real network. Consequently, we need to reconsider the weakness of NTK in the study of network theory. Also, the results show that mirrored initialization is superior to standard initialization in practical applications.

## 1.2 Related works

Our research is conducted within the framework of NTK theory. This type of research, in general, can be categorized into two main steps: the approximation of the network trained by GD through a kernel regression problem, and the evaluation on the corresponding kernel regression predictor. Several studies [2, 4, 19, 31] which focused on the former step, illustrated the point-wise convergence of NTK for multi-layer ReLU networks. Additionally, [39] demonstrated the point-wise convergence of the kernel regression predictor to the network. Furthermore, Lai et al. [35], Li et al. [41], Xu and Zhu [54] demonstrated the uniform convergence result with respect to all input and all time on two-layer and multi-layer networks. As to the latter step, a few researchers have analyzed the spectral properties of the NTK [6, 7] as well as kernel regression [40, 55]. Building upon these findings, Lai et al. [35] and Li et al. [41] demonstrated that early-stopping GD induces minimax optimality of the network. It is worth noting that the setting in these works assumes mirrored initialization, which may not be well-aligned with real-world scenarios. When it comes to initialization, Zhang et al. [56] provided insights into the impact of initialization under kernel interpolation, which is a special case of our results at $t = \infty$.

# 2 Preliminaries

## 2.1 Model and notations

Suppose that $\{(x_i, y_i)\}_{i=1}^n$ are i.i.d. drawn from an unknown distribution $\rho$ which is given by

$$y = f^*(x) + \epsilon, \tag{1}$$

where $f^*(x)$ is the *regression function* and $\epsilon$ is a centered random noise. Suppose that the marginal distribution $\mu(x)$ of the radom variable $x$ is supported in a non-empty bounded subset $\mathcal{X}$ of $\mathbb{R}^d$ with $C^\infty$ smooth boundary. The generalization error of an estimator $\hat{f}$ of $f^*$ is given by excess risk

$$\mathcal{E}(\hat{f}; f^*) = \left\| \hat{f} - f^* \right\|_{L_2(\mathcal{X}, \mu)}^2. \tag{2}$$

We introduce the following standard assumption on the noise(e.g., [21, 42]). It is clear that sub-Gaussian noise satisfying this assumption.

**Assumption 1** (Noise). The noise term $\epsilon$ satisfies the following condition for some positive constant $\sigma, L$, and $m \geq 2$:

$$\mathbf{E}(|\epsilon|^m | x) \leq \frac{1}{2} m! \sigma^2 L^{m-2}, \quad a.e.\ x \in \mathcal{X}. \tag{3}$$

**Notations** Given a set of samples pairs $\{(x_i, y_i)\}_{i=1}^n$, we denote $X$ and $Y$ to be vector $(x_1, \cdots, x_n)^T$ and $(y_1, \cdots, y_n)^T$, respectively. In a similar manner, $(f(x_1), \cdots, f(x_n))^T$ and $(f(y_1), \cdots, f(y_n))^T$ are represented as $f(X)$ and $f(Y)$, where $f(\cdot) : \mathbb{R}^d \mapsto \mathbb{R}$ is an arbitrary given function. Regarding a kernel function $k(\cdot, \cdot) : \mathbb{R}^d \times \mathbb{R}^d \mapsto \mathbb{R}$, we use $k(x, X)$ to denote the vector $(k(x, x_1), k(x, x_2), \cdots, k(x, x_n))$ and $k(X, X)$ to denote the matrix $[k(x_i, x_j)]_{n \times n}$. For real number sequences such as $\{a_n\}$ and $\{b_n\}$, we write $a_n = O(b_n)$ (or $a_n = o(b_n)$), if there exists absolute positive constant $C$ such that $|a_n| \leq C|b_n|$ holds for any sufficiently large $n$ (or $|a_n|/|b_n|$ approaches zero). We also denote $a_n \asymp b_n$ if there exists absolute positive constant $c$ abd $C$ such that $c|b_n| \leq |a_n| \leq C|b_n|$ holds for any sufficiently large $n$.

## 2.2 Reproducing kernel Hilbert space

Suppose that $k$ is kernel function defined on the domain $\mathcal{X}$ satisfying that $\|k\|_\infty \leq \kappa^2$. Let $\mathcal{H}_k$ be the reproducing Hilbert space associated with $k$ which is the closure of linear span of $\{k(x, \cdot), x \in \mathcal{X}\}$

under the inner product induced by $\langle k(x, \cdot), k(y, \cdot)\rangle = k(x, y)$. Given a distribution $\mu(x)$ on $\mathcal{X}$, we can introduce an integral operator $T_k : L^2(\mathcal{X}, \mu) \to L^2(\mathcal{X}, \mu)$:

$$T_k f(x) = \int_{\mathcal{X}} k(x, y) f(y) \, d\mu(y). \tag{4}$$

The celebrated Mercer's decomposition [12] asserts that

$$T_k f = \sum_{i \in \mathbb{N}} \lambda_i \langle f, e_i \rangle_{L^2} e_i, \quad k(x, y) = \sum_{i \in \mathbb{N}} \lambda_i e_i(x) e_i(y), \tag{5}$$

where $\{e_i\}_{i \in \mathbb{N}}$ and $\{\lambda_i^{\frac{1}{2}} e_i\}_{i \in \mathbb{N}}$ are the orthonormal basis of $L^2(\mathcal{X}, \mu)$ and $\mathcal{H}_k$ respectively. It is well known that $\mathcal{H}_k$ can be canonically embedded into $L^2(\mathcal{X}, \mu)$.

If the eigenvalues $\lambda_i$ of $k$ are polynomially decaying at rate $\beta$ ( i.e, $\lambda_i \asymp i^{-\beta}$), we can further introduce a concept of the relative smoothness of a function $f \in L^2(\mathcal{X}, \mu)$. More precisely, let us recall the concept of *real interpolation space* [48] ( Please see more detailed information in the Appendix).

**Real interpolation space**    The real interpolation space $[\mathcal{H}_k]^s$ is given by

$$[\mathcal{H}_k]^s := \left\{ \sum_{i \in \mathbb{N}} a_i \lambda_i^{\frac{s}{2}} e_i(x) \Big| \sum_{i \in \mathbb{N}} a_i^2 < \infty \right\}, \tag{6}$$

with the inner product $\langle \sum_{i \in \mathbb{N}} a_i \lambda_i^{\frac{s}{2}} e_i(x), \sum_{i \in \mathbb{N}} b_j \lambda_j^{\frac{s}{2}} e_j(x) \rangle_{[\mathcal{H}_k]^s} = \sum_{i \in \mathbb{N}} a_i b_i$ for $s \geq 0$.

It is clear that $[\mathcal{H}_k]^s$ is a separable Hilbert space and is isometric to the $l_2$ space. With the definition above, we can see that $[\mathcal{H}_k]^0 = L^2(\mathcal{X}, \mu)$ and $[\mathcal{H}_k]^1 = \mathcal{H}_k$. Also, for any $s_2 \geq s_1 \geq 0$, we know $[\mathcal{H}_k]^{s_1} \subseteq [\mathcal{H}_k]^{s_2}$ with compact embedding. Let

$$\alpha_0 = \inf_s \left\{ s \mid [\mathcal{H}_k]^s \subseteq C^0(\mathcal{X}) \right\}$$

which is often referred to the embedding index of an RKHS $\mathcal{H}_k$ [21] . It is well known that $\alpha_0 \geq \frac{1}{\beta}$ and the equality holds for a large class of usual RKHSs if the eigenvalue decay rate is $\beta$. We further define the relative smoothness of a given function $f$:

**Definition 2.1** (Relative smoothness). Given a kernel $k$ on $\mathcal{X}$ with respect to measure $\mu$, the smoothness of a function $f$ is defined as

$$\alpha(f, k) = \sup \left\{ \alpha > 0 \Big| \sum_{i \in \mathbb{N}} \lambda_i^{-\alpha} c_i^2 < \infty \right\}, \tag{7}$$

where $c_i = \langle f, e_i \rangle_{L^2(\mathcal{X}, \mu)}$.

## 2.3   Kernel gradient flow

For a positive definite reproducing kernel $k$, the dynamic of kernel gradient flow (KGF) [22] is

$$\frac{d}{dt} f_t^{\mathrm{GF}}(x) = -\frac{1}{n} k(x, X) \left( f_t^{\mathrm{GF}}(X) - Y \right), \tag{8}$$

where $f_t^{\mathrm{GF}}$ is the KGF predictor. In kernel gradient flow, the performance of kernel predictor depends on the relative smoothness of regression function. People often consider the case that $\alpha(f, k) \geq 1$ [10, 11] . When the smoothness satisfies $\alpha(f, k) < 1$, the regression function is said to be poorly smooth and belongs to the so-called misspecified spectral algorithm problem. We collect the related result in Zhang et al. [55] and apply it to our case, to derive the following proposition:

**Proposition 2.2.** Suppose the eigenvalue decay rate of $k$ is $\beta$ and the embedding index is $\frac{1}{\beta}$ with respect to $\mu$. Suppose the noise term $\epsilon$ satisfies Assumption 1. Let the dynamic (8) starts from $f_0^{\mathrm{GF}} = 0$. Also, suppose the regression function satisfies $f^* \in [\mathcal{H}_k]^s$ and $\|f^*\|_{[\mathcal{H}_k]^s} \leq R$, for some $s > 0$. Let $\gamma \leq s$ and $0 \leq \gamma \leq 1$. By choosing $t \asymp n^{\frac{\beta}{s\beta + 1}}$, for any fixed $\delta \in (0, 1)$, when $n$ is sufficient large, with probability at least $1 - \delta$, we have

$$\left\| f_t^{\mathrm{GF}} - f^* \right\|_{[\mathcal{H}_k]^\gamma}^2 \leq \left( \ln \frac{6}{\delta} \right)^2 R^2 C n^{-\frac{(s-\gamma)\beta}{s\beta + 1}},$$

where $C$ is a positive constant.

# 3 Network and Neural Tangent Kernel

## 3.1 Network settings

We consider the fully-connected network with $L$ hidden layers. As is commonly-used in deep learning, we consider the ReLU activation [44] defined by $\sigma(x) := \max(x, 0)$. Denote $f(\cdot; \theta) : \mathbb{R}^d \to \mathbb{R}$ as the network output function, where $\theta$ representing the column vector that all parameters flattened into. We can write the recursive structure of network as following:

$$
\begin{aligned}
\alpha^{(1)}(x) &= \sqrt{\frac{2}{m_1}} \left( W^{(0)} x + b^{(0)} \right); \\
\alpha^{(l)}(x) &= \sqrt{\frac{2}{m_l}} W^{(l-1)}(x) \sigma(\alpha^{(l-1)}(x)), \quad l = 2, 3, \cdots, L; \\
f(x; \theta) &= W^{(L)} \sigma(\alpha^{(L)}(x)),
\end{aligned}
\tag{9}
$$

The parameter matrix for the $l$-th layer is denoted as $W^{(l)}$. Their dimensions are of $m_{l+1} \times m_l$, where $m_l$ is the number of units in layer $l$ and $m_{l+1}$ is that of layer $l+1$. Also, the bias term of the first layer is denoted as $b^{(0)} \in \mathbb{R}^{m_1 \times 1}$. The setting of bias term is to make sure the positive definiteness of NTK [41]. We further assume that the number of units in each layer is at the same order while the width comes to infinity, as $cm \leq \min(m_1, \cdots, m_{L+1}) \leq \max(m_1, \cdots, m_{L+1}) \leq Cm$ where $c, C$ are some absolute positive constants.

**Standard initialization** At initialization, the parameters are randomly set as i.i.d. standard normal variables:

$$
W_{ij}^{(l)}, b_k^{(0)} \overset{\text{i.i.d.}}{\sim} \mathcal{N}(0, 1), \quad l = 0, 1, \cdots, L; \quad k = 1, \cdots, m_1.
\tag{10}
$$

**Remark 3.1** (Mirrored initialization). As to the *mirrored initialization* considered in [4, 35, 41], part of the network $f^{(1)}(\cdot; \theta_0^{(1)})$ undergoes standard initialization, while the other complicated corresponding part $f^{(2)}(\cdot; \theta_0^{(2)})$ holds the same structure as $f^{(1)}(\cdot; \theta_0^{(1)})$, with parameters initialized to the same values as $\theta_0^{(2)} = \theta_0^{(1)}$. Lastly, the neural network output function is defined as $f(\cdot; \theta_0) = \frac{\sqrt{2}}{2} \left( f^{(1)}(\cdot; \theta_0^{(1)}) - f^{(2)}(\cdot; \theta_0^{(2)}) \right)$. This setup ensures that $f(\cdot; \theta_0)$ is constantly zero.

The network is trained under the mean square loss function. If we suppose $\{(x_i, y_i)\}_{i=1}^n$ be the training data, then the loss function is specified as

$$
\mathcal{L}(\theta) = \frac{1}{2n} \sum_{i=1}^{n} (f(x_i; \theta) - y_i)^2.
\tag{11}
$$

For notational simplicity, we denote by $f_t^{\text{NN}}(x) = f(x; \theta_t)$. The training process for the network is performed by gradient flow, where the parameters are updated through the differential equation:

$$
\frac{\mathrm{d}}{\mathrm{d}t} \theta_t = -\partial_\theta \mathcal{L}(\theta) = -\frac{1}{n} [\partial_\theta f_t^{\text{NN}}(X)]^T (f_t^{\text{NN}}(X) - Y),
\tag{12}
$$

where $\partial_\theta f_t^{\text{NN}}(X)$ is a matrix with dimensions $n \times M$, with $M$ being the length of the parameter vector $\theta$. This matrix represents the gradient of the network output $f_t^{\text{NN}}(X)$ with respect to the parameters $\theta$ at time $t$. Incorporating the chain rule, we can formulate the gradient flow equation for the network function as follows:

$$
\frac{\mathrm{d}}{\mathrm{d}t} f_t^{\text{NN}}(x) = -\frac{1}{n} \partial_\theta f_t^{\text{NN}}(x) [\partial_\theta f_t^{\text{NN}}(X)]^T (f_t^{\text{NN}}(X) - Y).
\tag{13}
$$

## 3.2 Network at initialization

In order to state the properties of wide network with standard initialization, we need to introduce the concept of Gaussian process.

**Gaussian process**    Gaussian process is a stochastic process for which every finite collection of random variables follows a multivariate Gaussian distribution. Let $X$ be a Gaussian process with index $t \in T$. If the mean and covariance are given by the mean function $m$ and the positive definite kernel $k$ such that $\mathbf{E}[X(t)] = m(t)$ and $\mathrm{Cov}[X(t)X(t')] = k(t,t')$, which holds for any $t, t' \in T$, then we say $X \sim \mathcal{GP}(m, k)$.

In standard initialization (10), the parameters of the neural network are i.i.d. samples from a standard normal distribution. If the network contains only one hidden-layer (that is, if $L = 2$), it is direct to prove that $f_0^{\mathrm{NN}}(x)$ converges to a centered Gaussian distribution by CLT, for any fixed point $x \in \mathcal{X}$. As to the multi-layer network, prior research [25] also proved that such initialized network converges to a Gaussian process, as following:

**Lemma 3.2** (Limit distribution of initialization). *As the network width $m$ tend to infinity, the sequence of network stochastic process $\{f_0^{\mathrm{NN}}\}_{m=1}^{\infty}$ converges weakly in $C(\mathcal{X}, \mathbb{R})$ to a centered Gaussian process $f^{\mathrm{GP}}$. The covariance function is the so-called random feature kernel (RFK), which is denoted by $K^{\mathrm{RFK}}(x, x')$ as defined in (42) in Appendix C.1.*

### 3.3   The kernel regime

As the gradient descent of neural network involves high non-linearity and non-convexity, it is difficult to study the training process. However, Jacot et al. [31] introduced the Neural Tangent Kernel (NTK) theory which provides a connection between network training and a class of kernel regression problems, when the network width comes to infinity. To demonstrate this, we first define a Neural Network Kernel (NNK):

$$K_t^m(x, x') = [\partial_\theta f_t(x)]^T [\partial_\theta f_t(x')]. \tag{14}$$

Using this notation, we reformulate (13) in a kernel regression format:

$$\frac{\mathrm{d}}{\mathrm{d}t} f_t^{\mathrm{NN}}(x) = -\frac{1}{n} K_t^m(x, X)(f_t^{\mathrm{NN}}(X) - Y). \tag{15}$$

NTK theory shows, if the network width $m$ tends to infinity, then the random kernel $K_t^m(\cdot, \cdot)$ will converge to a time-invariant kernel $K^{\mathrm{NTK}}(\cdot, \cdot) : \mathbb{R}^d \times \mathbb{R}^d \to \mathbb{R}$, which is referred to as the NTK of network. The phenomenon is the so-called NTK regime [2, 31, 39]. The fixed kernel $K^{\mathrm{NTK}}$ only depends on the structure of the neural network and the way of initialization. To get more knowledge of NTK, we present the explicit expression of NTK in Appendix C.1. In NTK theory, the dynamic of network (15) can be approximated by a kernel gradient flow equation:

$$\frac{\mathrm{d}}{\mathrm{d}t} f_t^{\mathrm{NTK}}(x) = -\frac{1}{n} K^{\mathrm{NTK}}(x, X)(f_t^{\mathrm{NTK}}(X) - Y), \tag{16}$$

which starts from Gaussian process $f_0^{\mathrm{NTK}} = f^{\mathrm{GP}}$. In this way, if we aims to derive the generalization property of sufficiently wide network, we can achieve by considering the corresponding kernel gradient flow predictor. Such approximation is strictly ensured by uniform convergence of $f_t^{\mathrm{NN}}$ and $f_t^{\mathrm{NTK}}$ over all $x \in \mathcal{X}$ and all $t \geq 0$ as $m \to \infty$, since we use $L^2$ excess risk to evaluate the generalization ability. Actually, we have the following theorem, whose proof is given in Appendix B.

**Proposition 3.3** (Uniform convergence). Given training sample pairs $\{(x_i, y_i)\}_{i=1}^n$. For any $\delta \in (0, 1)$ and $\varepsilon > 0$, when network width $m$ is large enough, we have

$$\sup_{x, x' \in \mathcal{X}} \sup_{t \geq 0} |f_t^{\mathrm{NN}}(x) - f_t^{\mathrm{NTK}}(x)| \leq \varepsilon$$

holds with probability at least $1 - \delta$ .

In this theorem, we show the uniform convergence of network under standard initialization. Previous related studies [4, 35, 41, 54] always utilized delicately designed mirrored initialization (as shown in Remark 3.1) to avoid the analysis on the initial output function of network, since it will lead to the challenging problem that $f_t^{\mathrm{NN}}$ and $f_t^{\mathrm{NTK}}$ are both random, unlike that $f_t^{\mathrm{NTK}}$ is a fixed function in the case of mirrored initialization. However, as shown in Section 3.2, the initial output function is near a Gaussian process that can not be overlooked. To under the performance of neural networks commonly used in real world, it is necessary to analyzing the network initialization. To the best of our knowledge, we are the first to consider the initial output function of network in uniform convergence. This comprehensive result allows us to study the generalization error of network more precisely.

# 4 Impact of Initialization

## 4.1 Impact of standard initialization on the generalization error

The standard kernel gradient flow is always considered to start from zero, as in Proposition 2.2. Therefore, we need to do a transformation since the initial value of predictor $f_t^{\mathrm{NTK}}$ is actually $f^{\mathrm{GP}}$ instead of zero. Firstly, we can yield a solution of (8) in matrix form:

$$f_t^{\mathrm{GF}}(x) = f_0^{\mathrm{GF}}(x) + k(x, X)(I - e^{-\frac{1}{n}k(X,X)})\,[k(X,X)]^{-1}\,(f_0^{\mathrm{GF}}(X) - f^*(X) - \epsilon_X), \quad (17)$$

where $\epsilon_X$ is employed to represent the $n \times 1$ column noise term vector $Y - f^*(X)$. We denote by $f_t^{\mathrm{GF}}$ the kernel gradient flow predictor under initial function $f_0$ and denote by $\widetilde{f}_t^{\mathrm{GF}}$ the KGF predictor under initialization $\widetilde{f}_0^{\mathrm{GF}} \equiv 0$. If we plug them into (17) and excess risk (2), respectively, we directly have the following theorem:

**Proposition 4.1** (Impact of initialization in kernel gradient flow). Denote $\widetilde{f}^* = f^* - f_0$ as the biased regression function. For the KGF predictor $f_t^{\mathrm{GF}}$ and $\widetilde{f}_t^{\mathrm{GF}}$ defined above, we have

$$\mathcal{E}(f_t^{\mathrm{GF}}; f^*) = \mathcal{E}(\widetilde{f}_t^{\mathrm{GF}}; \widetilde{f}^*). \tag{18}$$

The theorem establishes the equivalence of the generalization properties between the KGF predictor with initial value $f_0$, regression function $f^*$ and the KGF predictor with initial value zero, regression function $f^* - f_0$. Back to the network case, combining uniform convergence result in Proposition 3.3, it suggests that, compared to mirrored initialization, the impact of standard initialization which has non-zero initial output function is equivalent to introducing a same-valued implicit bias to the regression function. This is a generalization of the main result in Zhang et al. [56], which only focused on case at $t = \infty$. To summarize, Proposition 4.1 provides a convenient approach to quantify the impact of standard initialization in early-stopping neural networks.

## 4.2 Smoothness of Gaussian process

Building upon the analysis above, our focus now turns to illustrating the smoothness of the Gaussian process $f^{\mathrm{GP}}$, as it is the limit distribution of $f_0^{\mathrm{NN}}$. Actually, we can derive the following theorem:

**Theorem 4.2** (Smoothness of Gaussian Process). *Suppose that $f^{\mathrm{GP}}$ is a Gaussian process with mean function 0 and covariance function $K^{\mathrm{RFK}}$. The following statements hold:*

$$\begin{aligned}
\mathbf{P}\left(f^{\mathrm{GP}} \in [\mathcal{H}^{\mathrm{NT}}]^s\right) = 1, \qquad s < \frac{3}{d+1}; \\
\mathbf{P}\left(f^{\mathrm{GP}} \in [\mathcal{H}^{\mathrm{NT}}]^s\right) = 0, \qquad s \geq \frac{3}{d+1}.
\end{aligned} \tag{19}$$

We furnish a comprehensive proof for Theorem 4.2 in the Appendix C.

Let us now turn our attention to the implications established by this theorem. Recall that Proposition 4.1 has shown that, in KGF, the existence of initialization function $f^{\mathrm{GP}}$ is equivalent to adding a same-valued bias term to the regression function $f^*$. Consequently, the poor smoothness of initialization function causes the high smoothness assumption on the regression function meaningless. Regardless of how smooth we assume the regression function to be (e.g., $\alpha(f^*, K^{\mathrm{NTK}}) \geq 2$), the value of (relavtive) smoothness $\alpha(f^* - f^{\mathrm{GP}}, K^{\mathrm{NTK}})$ will always be at most $\frac{3}{d+1}$. Namely, the biased regression function $f^* - f^{\mathrm{GP}}$ is always poorly smooth. In this specific case, we could hardly expect the KGF predictor to have fine performance.

## 4.3 Upper bound

Now we are ready to provide the upper bound of generalization error of network. With the help of Proposition 2.2, Proposition 3.3, Proposition 4.1 and Theorem 4.2, we derive the following theorem:

**Theorem 4.3** (Generalization error upper bound). *Assume that the regression function $f^* \in [\mathcal{H}^{\mathrm{NT}}]^s$ for some $s > 0$, and $\|f^*\|_{[\mathcal{H}^{\mathrm{NT}}]^s} \leq R$ where $R$ is a positive constant. Assume the marginal probability measure $\mu$ with density $p(x)$ satisfies $c \leq p(x) \leq C$ for some positive constant $c$ and $C$.*

- *For the case of $s \geq \frac{3}{d+1}$, for any $\delta \in (0,1)$ and $\varepsilon \in (0, \frac{3}{d+3})$, by choosing certain $t = t(n) \to \infty$ (as shown in Appendix), when $n$ is sufficiently large and $m$ is sufficiently large, with probability $1 - \delta$ we have*

$$\left\| f_t^{\mathrm{NN}} - f^* \right\|_{L^2}^2 \leq \left( \frac{1}{\delta} \ln \frac{6}{\delta} \right)^2 (R + C_\varepsilon)^2 C n^{-\frac{3}{d+3} + \varepsilon}, \tag{20}$$

  *where $C_\varepsilon$ is a positive constant related to $\varepsilon$.*

- *For the case of $0 < s < \frac{3}{d+1}$, for any $\delta \in (0,1)$, by choosing $t \asymp n^{\frac{d+1}{s(d+1)+d}}$, when $n$ is sufficiently large and $m$ is sufficiently large, with probability $1 - \delta$ we have*

$$\left\| f_t^{\mathrm{NN}} - f^* \right\|_{L^2}^2 \leq \left( \frac{1}{\delta} \ln \frac{6}{\delta} \right)^2 (R + C_s)^2 C n^{-\frac{s(d+1)}{s(d+1)+d}}, \tag{21}$$

  *where $C_s$ is a positive constant related to $s$.*

The proof is provided in Appendix D. This result shows the generalization error upper bound for network with standard initialization and demonstrate its negative effect. Even if the goal function $f^*$ is quite smooth, the generalization error upper bound $n^{-\frac{3}{d+3}}$ remains to be a quite low rate, particularly considering that the dimension $d$ of data is usually large in real world. It suggests that the network no longer generalizes well, even if we adopt the once useful early stopping strategy in Li et al. [41].

## 4.4 Lower bound

From the analysis above, we can see the poor generalization ability of network under standard initialization. Furthermore, in this section, we take spherical data as example and provide the lower bound of generalization error. Namely, we presume the input vectors $x$ are distributed on the sphere $\mathbb{S}^d$ with probability measure $\mu$, which is a common assumption in NTK theory [6, 31, 36, 53]. We also slightly change the network structure. Compared to the network (9), we eliminate the bias term of the initial layer, as shown in (40) in Appendix. In this case, the NTK of new network is denoted by $K_0^{\mathrm{NTK}}$, and the RKHS $\mathcal{H}_0^{\mathrm{NT}}(\mathbb{S}^d)$ is abbreviated as $\mathcal{H}_0^{\mathrm{NT}}$, whose detailed properties is also given in Appendix C.1. Additionally, we make more assumption on the noise of data. We assume the noise term $\epsilon$ in (1) to have a constant second moment, as $\mathbf{E}\left[|\epsilon|^2|x\right] = \sigma^2$ for $x \in \mathbb{S}^d, a.e..$ Under these conditions, with the help of method in Li et al. [40], we derive the theorem:

**Theorem 4.4** (Generalization error lower bound). *We assume that the regression function $f^* \in [\mathcal{H}_0^{\mathrm{NT}}]^s$ for some $s > \frac{3}{d+1}$, and denote by $\|f^*\|_{[\mathcal{H}_0^{\mathrm{NT}}]^s} \leq R$ where $R$ is a positive constant. Assume that $\mu$ is the uniform measure. For any $\delta \in (0,1)$, when $n$ is large enough and $m$ is large enough, for any choice of $t = t(n) \to \infty$, with probability at least $1 - \delta$ we have*

$$\mathbf{E}\left[ \left\| f_t^{\mathrm{NN}} - f^* \right\|_{L^2}^2 | X \right] = \Omega\left( n^{-\frac{3}{d+3}} \right). \tag{22}$$

The proof is given in Appendix E. Through Theorem 4.4, we derive $n^{-\frac{3}{d+3}}$ as the generalization lower bound of standardly random-initialized network in NTK theory, even if the regression function is quite smooth. The rate $n^{-\frac{3}{d+3}}$ means model suffers notably from data that has large dimension: If $d$ is relatively large, then this rate of convergence can be extremely slow. This is a manifestation of the curse of dimensionality. In fact, it contrasts with the fact that neural networks excel at high-dimensional problems. This contradiction underscores the limitation of NTK theory for interpreting network performance.

## 5 Experiments

Our numerical experiments are conducted in two aspects to fully understand the impact of standard initialization. First, we show the performance of standard initialized network is indeed worse than the mirrored initialized case, on the aspect of learning rate. The phenomenon is in line with our theoretical analysis. Second, the smoothness of regression function of real data is significantly larger than $\frac{3}{d+1}$, which suggest the bad effect of non-zero intial output function of standard initialization will indeed destroy the performance of network if NTK theory holds. It demonstrates the drawback of NTK theory through contradiction.

## 5.1 Artificial data

In the first experiment, we employ artificial data to show the negative effect of standard initialization on the generalization error of network. The detailed settings are shown in Appendix F.

**Learning rate of network under different initialization**    The experiments are conducted for both $d = 5$ and $d = 10$, contrasting network performance subject to mirrored and standard initialization strategies. We choose a relatively smooth goal function to emphasize the impact of initialization. Specifically, we use $m = 20n$, epoch $= 10n$, and the gradient learning rate $lr = 0.6$. The networks are made sufficiently wide to ensure the overparametrization assumption is met. Additionally, we implement the early-stopping strategy as mentioned in Theorem 4.3, that is, selecting the minimum loss across all epochs as the generalization error. Finally, we test the network's generalization error on different levels of sample size $n$, and plot the log value of the generalization error corresponding to $\log(n)$ as shown in Figure 1. As we expected, the points in Figure 1 fits a linear trend. Moreover, the figure highlights the difference in learning rate under different initialization methods. This aligns with our theoretical results.

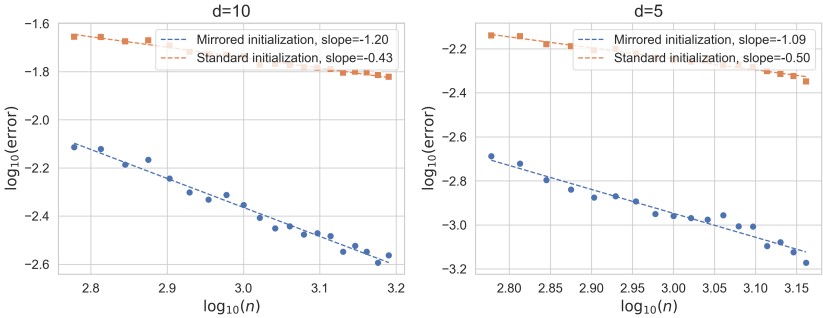

Figure 1: Generalization error decay curve of network. The scatter points show the averaged log error over $20$ trials. The dashed lines are computed through least-squares. The scale of $n$ is not broad because a larger $n$ requires a larger $m$, which would induce higher computational costs.

## 5.2 Real data

In this subsection, we focus on datasets from the real world and estimate the smoothness of function. Although we could not know the goal function that the real data is generated from, there exists a way to estimate its smoothness [16]. We show the technical details in Appendix G.

Table 1: Smoothness of goal function

| Dataset | | |
| --- | --- | --- |
| Name | Dimension | Smoothness |
| MNIST | $28 \times 28 \times 1$ | 0.40 |
| CIFAR-10 | $32 \times 32 \times 3$ | 0.09 |
| Fashion-MNIST | $28 \times 28 \times 1$ | 0.22 |

**Smoothness of goal function in real datasets**    We employed the MNIST, CIFAR-10 and Fashion-MNIST datasets[33, 38, 52]. In the experiments, we evaluate the smoothness of goal function of the datasets, with respect to the one-hidden layer NTK. The results are presented in Table 1. With the input dimension $d = 784, 3072, 784$, we can compute that the smoothness of initialization function is equal to $\frac{3}{d+1} \approx 0$. However, the smoothness of goal function is far better than $\frac{3}{d+1}$, which implies that standard initialization will indeed destroy the generalization performance, under NTK theory. The contradiction between NTK theory and the real situation shows its limitation and once again confirms our conclusion.

# 6 Discussion

To summarize, this research focuses on the impact of standard random initialization on generalization property of fully-connected network in the NTK theory, which makes up the gap in this field. Many previous work [35, 41] verified the statistical optimality of neural network under delicately designed mirrored initialization, whose initial output function of network is zero. However, through our study, we pinpoint that if we consider the commonly-used standard initialization, the learning rate of network is notably slow when the dimension of data is slightly large, which fails to explain network's favorable performance in overcoming the curse of dimensionality. A direct implication of our work is the superiority of mirror initialization over standard initialization, which suggests a direction for future improvements. On a deeper level, although NTK theory can describe many properties of network, at least for the fully connected networks with Gaussian initialization discussed in this paper, we can explore better theoretical frameworks to characterize their generalization ability in the future.

## Acknowledgments

Lin's research was supported in part by the National Natural Science Foundation of China (Grant 92370122, Grant 11971257).

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

# A  Further notations

In appendix, we will provide many technical proofs. Before that, let us provide more notations. For two sets $A$ and $B$ with a mapping function $\phi : A \to B$, the notation $\phi(A)$ is used to denote the image set of $A$ under $\phi$. For two random variable sequences $\{u_n\}$ and $\{v_n\}$, we denote by $u_n = o_{\mathbf{P}}(v_n)$ (or $u_n = \Omega_{\mathbf{P}}(v_n)$) if the ratio $u_n/v_n$ approaches zero (or $u_n \geq c v_n$ for some positive constant $c$) in probability as $n \to \infty$ with respect to probability measure $\mathbf{P}$. For two real number sequence $\{a_n\}$ and $\{b_n\}$, we denote by $a_n = \Omega(b_n)$ if there exists positive constant $c$ and $n_0$ such that $|a_n| \geq c|b_n|$ holds for any $n \geq n_0$. For two sequences of real numbers $\{a_n\}$ and $\{b_n\}$ such that $a_n = \Omega(b_n)$ (or $a_n = O(b_n)$), we also denote by $a_n \gtrsim b_n$ (or $a_n \lesssim b_n$). If $a_n \gtrsim b_n$ and $a_n \lesssim b_n$, then we denote by $a_n \asymp b_n$.

# B  Proof of uniform convergence

In this section, we demonstrate the uniform convergence from $f_t^{\mathrm{NN}}$ to $f_t^{\mathrm{NTK}}$.

## B.1  Initialization

The following is a direct proposition based on Lemma H.2 and Lemma 3.2,

**Proposition B.1.** For the random network function sequence $\{f_0^m\}$ with probability measures on $(C(\mathcal{X}, \mathbb{R}), \mathcal{C})$, there exists $\{X_m\}$ and $X^{\mathrm{GP}}$ defined on a new probability space $(\Omega', \mathcal{F}, \mathbf{P})$, on which we have
$$\mathbf{P}(\lim_{m \to \infty} \|X_m - X^{\mathrm{GP}}\|_\infty = 0) = 1.$$
where $X_m$ and $X^{\mathrm{GP}}$ has the same distribution as $f_0^m$ and $f^{\mathrm{GP}}$, respectively.

**Remark B.2.** The separability of $(C(\mathcal{X}, \mathbb{R}), \mathcal{C})$ can be derived by the density of polynomials. Therefore, it satisfies the requirement of Lemma H.2. In the context of our study, our reliance is only on the distribution of $\{f_0^m\}$ for each given value of $m$. Consequently, it is reasonable to reconstruct it in the new probability space. For convenience, we directly denote $X_0^m$ as $f_0^m$ (or $f_0^{\mathrm{NN}}$) and denote and $X^{\mathrm{GP}}$ as $f^{\mathrm{GP}}$, respectively. In other words, we are considering the network function in a new probability space, even though this approach may result in a moderate abuse of notation.

## B.2  Uniform convergence of network

Our aim is to give the uniform convergence between NTK regressor $f_t^{\mathrm{NTK}}$ and network function $f_t^{\mathrm{NN}}$. Note that the NTK regressor is trained by NTK, and the network function is trained by NNK, which is denoted by $K_t^m$. Here we first show the uniform convergence between NNK and NTK as $m$ comes to infinity.

**Lemma B.3.** *For any $\delta \in (0,1)$, suppose $m$ is large enough, then with probability at least $1 - \delta$, we have*
$$\sup_{t \geq 0} \sup_{x, x' \in \mathcal{X}} |K_t^m(x, x') - K^{\mathrm{NTK}}(x, x')| \leq O(m^{-\frac{1}{12}} \sqrt{\log m}).$$

*Proof.* The proof is similar to that in Li et al. [41], while the difference is the way of initialization. So we only provide the sketch of proof. In Li et al. [41], the uniform convergence of NTK is proved through a standard $\epsilon - net$ argument, which is divided into point-wise convergence and continuity of both NTK and NNK. Namely, as the following decomposition:
$$\begin{aligned} |K_t^m(x, x') - K^{\mathrm{NTK}}(x, x')| \leq &|K_t^m(x, x') - K_t^m(z, z')| \\ &+ |K_t^m(z, z') - K^{\mathrm{NTK}}(z, z')| + |K^{\mathrm{NTK}}(z, z') - K^{\mathrm{NTK}}(x, x')|. \end{aligned} \tag{23}$$
where $z, z'$ are the points in the $\epsilon - net$ which divides $\mathcal{X}$.

Back to our case, in non-zero initialization, the structrue of NTK and NNK remain the same, as well as the continuity property. Consequently, the effect of initialization reflects on the point-wise convergence from $K_t^m(z, z')$ to $K^{\mathrm{NTK}}(z, z')$, or more precisely, the NTK regime [2]. NTK regime requires that the residual decays to near zero and thereby the parameters will not deviate too far from their initial values in the training process, which holds under mirrored initialization. Standard

initialization lets the residual at time 0 be $\left\| f_0^{\mathrm{NN}}(X) - Y \right\|_2$, instead of $\|Y\|_2$. Therefore, there is a slight risk that the residual is too large to decay to near zero during training. However, since

$$\left\| f_0^{\mathrm{NN}}(X) \right\|_2 \leq O(n \cdot m^{\frac{1}{8}}), \tag{24}$$

holds with high probability when $m$ is large through Proposition B.1 and direct analysis on $f^{\mathrm{GP}}$, we can verify that the residual $\left\| f_t^{\mathrm{NN}}(X) - Y \right\|_2$ is still not large enough to break the stable lazy regime. Namely, the control on parameter matrix that

$$\sup_{t \geq 0} \left\| W_t^{(l)} - W_0^{(l)} \right\|_{\mathrm{F}} = O(m^{\frac{1}{4}}). \tag{25}$$

still holds. In this way, we can finish the proof. $\qquad \square$

Then, we can derive the uniform convergence of network function.

*Proof of Proposition 3.3.* The proof is also similar to that of uniform convergence under mirrored initialization. Therefore, we only exhibit the sketch of different part. Define event $A$ as

$$A = \left\{ \|f_0^{\mathrm{NN}} - f^{\mathrm{GP}}\|_\infty \leq o_m(1) \right\} \cap \left\{ \left\| f^{\mathrm{GP}}(X) \right\|_2 \leq C_\delta \right\} \tag{26}$$

where $C_\delta$ is some constant related to $\delta$, such that event $A$ holds with probability at least $1 - \frac{\delta}{2}$ when $m$ is large enough. Such a constant $C_\delta$ is ascertainable, as $f_0^{\mathrm{NN}}$ converges to $f^{\mathrm{GP}}$ by Proposition B.1 and $f^{\mathrm{GP}}$ is a Gaussian process with finite second moment. Define event $B$ as

$$B = \left\{ \sup_{t \geq 0} \sup_{x, x' \in \mathcal{X}} |K_t^m(x, x') - K^{\mathrm{NTK}}(x, x')| \leq o_m(1) \right\}. \tag{27}$$

We have event $B$ holds with probability at least $1 - \frac{\delta}{2}$ when $m$ is large enough. Conditioned on event $A$ and $B$, we do kernel gradient flow by $K_t^m$ and $K^{\mathrm{NTK}}$ on $f_0^{\mathrm{NN}}$ and $f^{\mathrm{GP}}$ respectively. Let event $C$ be

$$C = \left\{ \sup_{t \geq 0} \|f_t^{\mathrm{NN}} - f_t^{\mathrm{NTK}}\|_\infty \leq o_m(1) \right\}. \tag{28}$$

Conditioned on event $A$ and $B$, we can prove that event $C$ holds by Gronwall's inequality, as the same method in Lai et al. [35]. In this way, we can finish the proof. $\qquad \square$

After we get the uniform convergence of network function, we can obtain the proposition on the convergence of excess risk:

**Proposition B.4.** Suppose $f^* \in L^2(\mathcal{X}, \mu)$. For any $\delta \in (0, 1)$ and $\varepsilon > 0$, when $m$ is large enough , with probability at least $1 - \delta$, we have

$$\sup_{t > 0} \left| \left\| f_t^{\mathrm{NN}} - f^* \right\|_{L^2}^2 - \left\| f_t^{\mathrm{NTK}} - f^* \right\|_{L^2}^2 \right| \leq \varepsilon \tag{29}$$

*Proof.* Recall the dynamic equation of $f_t^{\mathrm{NTK}}$, we have

$$|f_t^{\mathrm{NTK}}(x)| \leq \left\| K^{\mathrm{NTK}}(x, X)^T \right\|_2 \left\| K^{\mathrm{NTK}}(X, X)^{-1} \right\|_2 \left\| f_0^{\mathrm{NTK}}(X) - Y \right\|_2. \tag{30}$$

Since the kernel function $K^{\mathrm{NTK}}(\cdot, \cdot)$ is bounded, there exists some positive constant $C$, such that

$$\left\| K^{\mathrm{NTK}}(x, X)^T \right\|_2 \leq C\sqrt{n}. \tag{31}$$

The initial function of kernel gradient flow $f_0^{\mathrm{NTK}} = f^{\mathrm{GP}}$ follows a Gaussian process with mean 0 and covariance kernel function $K^{\mathrm{RFK}}$. By the boundness of $K^{\mathrm{RFK}}$, we can also bound $f_0^{\mathrm{NTK}}$. That is, for any $\delta \in (0, 1)$, there exists a positive constant $M_\delta$ such that with probability at least $1 - \delta/2$,

$$\left\| f_0^{\mathrm{NTK}}(X) \right\|_2 \leq \sqrt{n} M_\delta. \tag{32}$$

Denote $\lambda_0 := \lambda_{\min}\left(K^{\mathrm{NTK}}(X, X)\right)$. We have $\lambda > 0$ since $K^{\mathrm{NTK}}$ is strictly positive definite [41]. Thus we have

$$|f_t^{\mathrm{NTK}}(x)| \leq C\sqrt{n} \lambda_0^{-1} (\sqrt{n} M_\delta + \|Y\|_2). \tag{33}$$

The excess risk

$$\left|\mathcal{E}(f_t^{\mathrm{NN}}; f^*) - \mathcal{E}(f_t^{\mathrm{NTK}}; f^*)\right| = \left|\int_{\mathcal{X}} |f_t^{\mathrm{NN}} - f_t^{\mathrm{NTK}}|^2 \, \mathrm{d}\mu + \int_{\mathcal{X}} (f_t^{\mathrm{NTK}} - f^*)(f_t^{\mathrm{NN}} - f_t^{\mathrm{NTK}}) \, \mathrm{d}\mu\right| \tag{34}$$

Since $f^* \in L^2(\mathcal{X}, \mu)$ where $\mu$ is probability measure, we also have $f^* \in L^1(\mathcal{X}, \mu)$. Denote $M_{f^*} := \|f^*\|_{L^1}$ and $\Delta := \sup_{x \in \mathcal{X}, t \geq 0} |f_t^{\mathrm{NN}}(x) - f_t^{\mathrm{NTK}}|$. We have

$$\left|\mathcal{E}(f_t^{\mathrm{NN}}; f^*) - \mathcal{E}(f_t^{\mathrm{NTK}}; f^*)\right| \leq \Delta^2 \cdot (1 + C\sqrt{n}\lambda_0^{-1}(\sqrt{n}M_\delta + \|Y\|_2) + M_{f^*}) \tag{35}$$

By Proposition 3.3, when $m$ is large enough, with probability at least $1 - \delta$ we have $\Delta^2 \leq \varepsilon/(1 + C\sqrt{n}\lambda_0^{-1}(\sqrt{n}M_\delta + \|Y\|_2) + M_{f^*})$. Thus the proposition is proved.

$\square$

# C  Proof of the Theorem 4.2

Before the proof, first we introduce some basic properties of NTK and RFK, as well as some technical properties of Sobolev space. We say that two Hilbert space $\mathcal{H}_1, \mathcal{H}_2$ are equivalent if they are equal as sets and share equivalent norm. If $\mathcal{H}_1$ and $\mathcal{H}_2$ are equivalent, we denote by $\mathcal{H}_1 \cong \mathcal{H}_2$.

## C.1  Basic properties of NTK and RFK

**Dot-product kernel**    A reproducing kernel function $k$ is dot-product if its value only depends on the dot-product of inputs. That is, there exists function $\kappa$ such that

$$k(x, x') = \kappa(\langle x, x' \rangle). \tag{36}$$

A dot-product kernel on sphere can be decomposed with spherical harmonic polynomials as the eigenfunction:

$$k(x, y) = \sum_{n=0}^{\infty} \mu_n \sum_{l=1}^{a_n} Y_{n,l}(x) Y_{n,l}(y). \tag{37}$$

where spherical harmonic polynomials $\{Y_{n,l}, l = 1, \cdots, a_n\}$ are also the orthonormal basis of $L^2(\mathbb{S}^d, \sigma)$, with $\sigma$ denoting the uniform measure on $\mathbb{S}^d$ [47]. This is also its Mercer decomposition.

Now come back to our network case. We first define two dot-product kernels on $\mathbb{S}^d$,

$$K_0^{\mathrm{NTK}}(x, y) := \sum_{r=0}^{L} \kappa_1^{(r)}(u) \prod_{s=r}^{L-1} \kappa_0(\kappa_1^{(s)}(u)), \quad K_0^{\mathrm{RFK}}(x, y) := \kappa_1^{(L)}(u), \tag{38}$$

where $u = \langle x, y \rangle = x^T y$ and

$$\kappa_0(u) = \frac{1}{\pi}(\pi - \arccos u), \qquad \kappa_1(u) = \frac{1}{\pi}\sqrt{1 - u^2} + \frac{u}{\pi}(\pi - \arccos u). \tag{39}$$

The definition of $\kappa_1^{(t)}$ is given by the composition $\kappa_1 \circ \kappa_1 \cdots \circ \kappa_1$ (a total of $t$ compositions). The explicit expression indicates that $K_0^{\mathrm{NTK}}$ and $K_0^{\mathrm{RFK}}$ are dot-product kernels on $\mathbb{S}^d$.

$K_0^{\mathrm{NTK}}$ and $K_0^{\mathrm{RFK}}$ is the homogeneous NTK and RFK of a homogeneous fully-connected network $f^S$ defined on $\mathbb{S}^d$ [6, 14], whose structural difference from (9) is the removal of the bias term in the first layer. Specifically, the network is structured as follows:

**Homogeneous fully-connected network on sphere**    The network is constructed using the following recursive formula:

$$\begin{aligned}
\alpha^{(1)}(x) &= \sqrt{\frac{2}{m_1}} W^{(0)} x; \\
\alpha^{(l)}(x) &= \sqrt{\frac{2}{m_l}} W^{(l-1)}(x) \sigma(\alpha^{(l-1)}(x)), \quad l = 2, 3, \cdots, L; \\
f^S(x; \theta) &= W^{(L)} \sigma(\alpha^{(L)}(x)),
\end{aligned} \tag{40}$$

where the function $\sigma$ is entrywise ReLU activation. The parameter matrix for the $l$-th layer is denoted as $W^{(l)}$, whose dimensions are of $m_{l+1} \times m_l$, where $m_l$ is the number of units in layer $l$ and $m_{l+1}$ is that of layer $l + 1$ for $l \in \{0, 1, \cdots, L - 1\}$. We also set $m_0$ to be equal to $d + 1$ and $m_{L+1}$ equal to 1. The network is also random initialized as (10).

We can easily build a connection between our NTK and RFK for network (9) and the homogeneous kernels defined in (38). For $x \in \mathcal{X}$, let $\widetilde{x} = (x, 1)$ which means add 1 as the new last component of $x$. Define $\phi(x) := \frac{(x,1)}{\|(x,1)\|}$ being an isomorphism from open set $\mathcal{X}$ to a subdomain of positive hemisphere shell $S = \phi(\mathcal{X}) \subset \mathbb{S}_+^d$. Then we have

$$f(x) = \|\widetilde{x}\| f^S(\phi(x)), \tag{41}$$

where $f$ is network (9) and $f^S$ is network (40). Actually, we can thus verify that

$$K^{\mathrm{NTK}}(x, y) = \|\widetilde{x}\| \|\widetilde{y}\| K_0^{\mathrm{NTK}}(\phi(x), \phi(y)), \qquad K^{\mathrm{RFK}}(x, y) = \|\widetilde{x}\| \|\widetilde{y}\| K_0^{\mathrm{RFK}}(\phi(x), \phi(y)). \tag{42}$$

We denote by $\mathcal{H}_0^{\mathrm{NT}}$ and $\mathcal{H}_0^{\mathrm{RF}}$ the RKHS on $\mathbb{S}^d$ with respect to $K_0^{\mathrm{NTK}}$ and $K_0^{\mathrm{RFK}}$. Their eigenvalue decay rates are well known:

**Lemma C.1** (Bietti and Bach [6], Haas et al. [24]). *For $K_0^{\mathrm{NTK}}$ and $K_0^{\mathrm{RFK}}$ on $\mathbb{S}^d$ with uniform measure $\sigma$, the decay rate of spherical harmonics coefficients satisfy*

$$\mu_n(K_0^{\mathrm{NTK}}) \asymp n^{-(d+1)} \quad and \quad \mu_n(K_0^{\mathrm{RFK}}) \asymp n^{-(d+3)}, \tag{43}$$

*while the eigenvalues satisfy*

$$\lambda_i(K_0^{\mathrm{NTK}}, \mathbb{S}^d, \sigma) \asymp i^{-\frac{d+1}{d}} \quad and \quad \lambda_i(K_0^{\mathrm{RFK}}, \mathbb{S}^d, \sigma) \asymp i^{\frac{d+3}{d}}. \tag{44}$$

Additionally, we list some further result on the eigenvalue decay rate of NTK and RFK provided by Li et al. [41], which will be used later:

**Lemma C.2.** *Denote $\Omega$ as a non-empty subdomain of $\mathbb{S}^d$. For $K_0^{\mathrm{NTK}}$ and $K_0^{\mathrm{RFK}}$, we have eigenvalue decay rate:*

$$\lambda_i(K_0^{\mathrm{NTK}}, \Omega, \sigma) \asymp i^{-\frac{d+1}{d}} \quad and \quad \lambda_i(K_0^{\mathrm{RFK}}, \Omega, \sigma) \asymp i^{-\frac{d+3}{d}}$$

*where $\sigma$ is the uniform measure on $S$.*

**Lemma C.3.** *For $K^{\mathrm{NTK}}$ and $K^{\mathrm{RFK}}$, we have eigenvalue decay rate:*

$$\lambda_i(K^{\mathrm{NTK}}, \mathcal{X}, \mu) \asymp i^{-\frac{d+1}{d}} \quad and \quad \lambda_i(K^{\mathrm{RFK}}, \mathcal{X}, \mu) \asymp i^{-\frac{d+3}{d}}$$

*where measure $\mu$ on bounded domain $\mathcal{X} \subset \mathbb{R}^d$ has density $c \leq p(x) \leq C$ with respect to the Lebesgue measure.*

## C.2 Basic concepts of Sobolev space

**Sobolev Space of integer power** Let $\mathcal{X}$ be a open subset of $\mathbb{R}^d$. Let $m \in \mathbb{N}$, $1 \leq p \leq +\infty$. Sobolev space $W^{m,p}(\mathcal{X})$ is defined as a set of function such that

$$\|D^\alpha f\|_{L^p} < +\infty, \tag{45}$$

where $\alpha$ is a vector with length $n$ and $D^\alpha f$ is the weak $\alpha$-th partial derivative of $f$. In other words, the definition of $W^{m,p}$ is:

$$W^{m,p}(\mathcal{X}) = \left\{ f \in L^p(\mathcal{X}) | D^\alpha f \in L^p(\mathcal{X}), \forall |\alpha| \leq m \right\}, \tag{46}$$

where $1 \leq p \leq \infty$. Conventionally, when the index $p$ is equal to 2, we denote $W^{m,p}$ by $H^m$, since it is a Hilbert space. Further, if the index $m > \frac{d}{2}$, the Sobolev space $H^m$ qualifies as a RKHS and thus embraces the properties of RKHS. In our work, we mainly utilize its property of interpolation as defined in (6). Consequently, we first introduce a generalized concept of real interpolation [1], as an expansion to the definition in (6).

**Real interpolation**    For two Bananch spaces $\mathcal{H}_1$ and $\mathcal{H}_2$, we use interpolation space to represent a space that lies in between them in some specific way. We introduce the commonly-used K-method to define real interpolation. Suppose $0 < s < 1, q \geq 1$. The space generated by their real interpolation $\mathcal{H} = (\mathcal{H}_1, \mathcal{H}_2)_{s,q}$ is defined by following:

$$K(t; x) = \inf_{x = x_1 + x_2; x_1 \in \mathcal{H}_1, x_2 \in \mathcal{H}_2} \|x_1\|_{\mathcal{H}_1} + t\|x_2\|_{\mathcal{H}_2}, \tag{47}$$

and

$$\|x\|_{\mathcal{H}} = \left( \int_0^\infty (t^{-s} K(t; x))^q \frac{\mathrm{d}t}{t} \right)^{\frac{1}{q}}. \tag{48}$$

Based on the definition of real interpolation, we introduce some basic concepts about fractional power Sobolev space.

**Sobolev space of fractional power**    Suppose $\mathcal{X} \in \mathbb{R}^d$ is a bounded domain with smooth boundary and denote Lebesgue measure by $\mu$. We can define fractional power Sobolev space through real interpolation (we refer to [45] Chapter 4.2.2 for more details):

$$H^s(\mathcal{X}) \coloneqq \left( L^2(\mathcal{X}, \mu), H^m(\mathcal{X}) \right)_{\frac{s}{m}, 2} \tag{49}$$

The fractional power Sobolev space $H^r(\mathcal{X})$ with $r \geq \frac{d}{2}$ is also a RKHS [1]. Specifically, Steinwart and Scovel [48] reveals that for $0 < s < 1$,

$$[\mathcal{H}]^s \cong \left( L^2(\mathcal{X}, \mu), \mathcal{H} \right)_{s, 2} \tag{50}$$

for RKHS $\mathcal{H}$ and the interpolation defined in (6). Therefore, the results above directly implies that

$$[H^r(\mathcal{X})]^s = H^{rs}(\mathcal{X}) \tag{51}$$

holds for any $r \geq \frac{d}{2}$ and $s > 0$.

Up to now, we have introduced the basic properties of Sobolev spaces on $\mathcal{X}$, an open subset of $\mathbb{R}^d$. For Sobolev spaces defined on more intricate manifolds, such as hyperspheres, owing to the intricate property of Sobolev spaces, numerous equivalent definitions emerges [1, 18].

We now delineate a kind of definition that will facilitate our subsequent proofs, since we will consider RKHSs on $\mathbb{S}^d$, like $\mathcal{H}_0^{\mathrm{NT}}$ and $\mathcal{H}_0^{\mathrm{RF}}$, which are the RKHSs associated with $K_0^{\mathrm{NTK}}$ and $K_0^{\mathrm{RFK}}$, respectively. Such definition can form a linkage with the Sobolev spaces defined on $\mathbb{S}^d$ and on domain $\mathcal{X} \subset \mathbb{R}^d$, which is also utilized in Haas et al. [24]. Our exposition begins with the characterization of a manifold.

**Trivilization**    Define a trivialization of a Riemannian manifold $(M, g)$ with bounded geometry of dimension $d$, which consists three part. The first part is some locally finite open covering $\{U_\alpha\}_{\alpha \in I}$. The second part is the charts $\{\kappa_\alpha\}_{\alpha \in I}$ which consists of smooth diffeomorphism $\kappa_\alpha : V_\alpha \subset \mathbb{R}^d \to U_\alpha$. The third part is a partion of unity $h_\alpha$ such that $\mathrm{supp}(h_\alpha) \subset U_\alpha, \sum_{\alpha \in I} h_\alpha = 1$ and $0 \leq h_\alpha \leq 1$.

In our case, we write a trivialization of $\mathbb{S}^d$, which, is a manifold of dimension $d$. We write $U_1 = \{x_{d+1} < \epsilon | x \in \mathbb{S}^d\}$ and $U_2 = \{x_{d+1} > \frac{\epsilon}{2} | x \in \mathbb{S}^d\}$ for a small fixed $\epsilon > 0$. Let $\phi_1 : U_1 \to \mathbb{R}^d$ and $\phi_2 : U_2 \to \mathbb{R}^d$ be stereographic projections with respect to $x_1 = (0, 0, \cdots, 1)$ and $x_2 = (0, 0, \cdots, -1)$, respectively. Namely, they are

$$\phi_1 : (x_1, x_2, \cdots, x_{d+1}) \mapsto \frac{1}{1 + x_{d+1}} (x_1, x_2, \cdots, x_d) \tag{52}$$

and

$$\phi_2 : (x_1, x_2, \cdots, x_{d+1}) \mapsto \frac{1}{1 - x_{d+1}} (x_1, x_2, \cdots, x_d). \tag{53}$$

Finally, we can find $C^\infty$ smooth functions $h_1$ and $h_2$ such that $h_1|_{\mathbb{S}^d_+} = 1$. For the simple trivialization above, we can directly verify that it meets the admissible trivialization condition (details see Große and Schneider [23]). Thus we can apply Theorem 14 of [23] to define the norm of Sobolev space on $\mathbb{S}^d$:

$$\|f\|_{H^s(\mathbb{S}^d)} = \left( \left\| (h_1 f) \circ \phi_1^{-1} \right\|_{H^s(\mathbb{R}^d)}^2 + \left\| (h_2 f) \circ \phi_2^{-1} \right\|_{H^s(\mathbb{R}^d)}^2 \right)^{\frac{1}{2}}, \tag{54}$$

for distribution $f \in \mathcal{D}'(\mathbb{S}^d)$ [49]. It gives a kind of equivalent definition of Sobolev space on $\mathbb{S}^d$.

## C.3  Relationship between dot-product kernel and Sobolev space

Previous work observed that, for dot-product kernels defined on sphere with polynomial eigenvalue decay rate, their RKHSs are equivalent to Sobolev spaces:

**Lemma C.4** (Hubbert et al. [30] Section 3). *For a dot-product kernel $k$ defined on $\mathbb{S}^d$ and its RKHS $\mathcal{H}_k$, if the coefficients of spherical harmonic polynomials satisfies $\mu_n \asymp n^t$ for some $t \geq d$, then there exists an equivalence between RKHS and Sobolev space:*

$$\mathcal{H}_k \cong H^{\frac{t}{2}}(\mathbb{S}^d).$$

Recall that $K_0^{\mathrm{NTK}}$ and $K_0^{\mathrm{RFK}}$ are both dot-product kernels with polynomial eigenvalue decay rate by Lemma C.1. Therefore, Lemma C.4 provides the equivalence between $\mathcal{H}_0^{\mathrm{NT}}$, $\mathcal{H}_0^{\mathrm{RF}}$ and the corresponding Sobolev spaces on $\mathbb{S}^d$. We have the following proposition:

**Proposition C.5.** We have the following equivalence:

$$\mathcal{H}_0^{\mathrm{NT}} \cong H^{\frac{d+1}{2}}(\mathbb{S}^d) \quad \text{and} \quad \mathcal{H}_0^{\mathrm{RF}} \cong H^{\frac{d+3}{2}}(\mathbb{S}^d). \tag{55}$$

## C.4  Interpolation of $\mathcal{H}_0^{\mathrm{NT}}$ and $\mathcal{H}_0^{\mathrm{RF}}$

In this subsection, we aims to provide the interpolation relationship between RKHSs associated with $K_0^{\mathrm{NTK}}$ and $K_0^{\mathrm{RFK}}$, on a subdomain of $\mathbb{S}^d$. We remind that if we consider the case on $\mathbb{S}^d$, i.e. $\mathcal{H}_0^{\mathrm{NT}}$ and $\mathcal{H}_0^{\mathrm{RF}}$, the conclusion is direct since they are both dot-product kernels and share the same orthogonal basis $\{Y_{n,l}\}$ as introduced in (37).

Suppose $s \geq \frac{d}{2}$ and $\Omega$ be a subdomain of $\mathbb{S}^d$ with $C^\infty$ smooth boundary. With a little abuse of notation, we define $H^s(\Omega)$ as the RKHS $H^s(\mathbb{S}^d)$ restricted to $\Omega$ in the way of Lemma H.3. For an injection $\varphi : \Omega \to \mathbb{R}^d$, we define $H^s(\varphi(\Omega)) \circ \varphi := \{f \circ \varphi | f \in H^s(\varphi(\Omega))\}$ with norm $\|f \circ \varphi\| = \|f\|_{H^s(\varphi(\Omega))}$. Recall that $\phi_1$ is the stereographic projection defined in (52), now let us show the equivalence of $H^s(\mathbb{S}_+^d)$ and $H^s(\phi_1(\mathbb{S}_+^d)) \circ \phi_1$.

**Lemma C.6** (Equivalence as sets). *Suppose $s \geq \frac{d}{2}$. At the aspect of sets, we have $H^s(\mathbb{S}_+^d) = H^s(\phi_1(\mathbb{S}_+^d)) \circ \phi_1$.*

*Proof.* For a $f \in H^s(\mathbb{S}_+^d)$, we have an extension $\|f'\|_{H^s(\mathbb{S}^d)} < \infty$ such $f'|_{\mathbb{S}_+^d} = f$. Thus

$$\left\|(h_1 f') \circ \phi_1^{-1}\right\|_{H^s(\mathbb{R}^d)} \leq \left\|f \circ \phi_1^{-1}\right\|_{H^s(\mathbb{R}^d)} < \infty \tag{56}$$

which implies $(h_1 f') \circ \phi_1^{-1} \in H^s(\mathbb{R}^d)$. Then we have $[(h_1 f') \circ \phi_1^{-1}]|_{\phi_1(\mathbb{S}_+^d)} \in H^s(\phi_1(\mathbb{S}_+^d))$. Since $f = f'|_{\mathbb{S}_+^d}$ and $h_1|_{\mathbb{S}_+^d} = 1$, we have $f \circ \phi_1^{-1} \in H^s(\phi_1(\mathbb{S}_+^d))$.

In the converse direction, we assume $f \in H^s(\phi_1(\mathbb{S}_+^d))$. Then we know there exists $f' \in H^s(\mathbb{R}^d)$ such that $f'|_{\phi_1(\mathbb{S}_+^d)} = f$. Now we want to show $f \circ \phi_1 \in H^s(\mathbb{S}_+^d)$. Define a $\psi \in C^\infty(\mathbb{R}^d)$ such that $\psi(\phi_1(\mathbb{S}_+^d)) \equiv 1$ and $\psi((\phi_1(U_1/U_2))^c) \equiv 0$. According to (54), we have

$$\|f \circ \phi_1\|_{H^s(\mathbb{S}_+^d)} \leq \|(\psi \cdot f') \circ \phi_1\|_{H^s(\mathbb{S}^d)} = \left\|(h_1 \circ \phi_1^{-1}) \cdot \psi \cdot f'\right\|_{H^s(\mathbb{R}^d)} < \infty \tag{57}$$

Thus we finish the proof. $\square$

**Lemma C.7** (Equivalence as space). *Suppose $s \geq \frac{d}{2}$. At the aspect of spaces, we have $H^s(\mathbb{S}_+^d) \cong H^s(\phi_1(\mathbb{S}_+^d)) \circ \phi_1$.*

*Proof.* By Lemma C.6, we know $H^s(\mathbb{S}_+^d) \cong H^s(\phi_1(\mathbb{S}_+^d)) \circ \phi_1$ as sets. Since $H^s(\mathbb{S}_+^d)$ and $H^s(\phi_1(\mathbb{S}_+^d)) \circ \phi_1$ are both RKHSs, we can finish the proof by closed graph theorem.

For notational simplicity, denote by $\mathcal{H}_1 = H^s(\mathbb{S}_+^d)$ and $\mathcal{H}_2 = H^s(\phi_1(\mathbb{S}_+^d)) \circ \phi_1$. Define the canonical map $I : \mathcal{H}_1 \to \mathcal{H}_2$ as $I : h \mapsto h$. Let $\{h_n\}_{n \in \mathbb{N}}$ be a sequence such that there exists $h \in \mathcal{H}_1$ and $g \in \mathcal{H}_2$ where $h_n \to h$ in $\mathcal{H}_1$ and $h_n = I h_n \to g$ in $\mathcal{H}_2$. It implies that $h = g$. Therefore, closed graph theorem shows that the linear operator $I$ is bounded, which means that $\|h\|_{\mathcal{H}_1} \leq C \|h\|_{\mathcal{H}_2}$ holds for some positive constant $C$ and any $h \in \mathcal{H}_1$. We can also prove $\|h\|_{\mathcal{H}_2} \leq C' \|h\|_{\mathcal{H}_1}$ for any $h$ in the same way. Consequently, the lemma is proved. $\square$

Now we come back to our network case. Let $S := \phi(\mathcal{X}) \subset \mathbb{S}_+^d$ where $\mathcal{X}$ is the set from which data $x$ is sampled, and $\phi$ is used in (42). Since the boundary of $\mathcal{X}$ is $C^\infty$ smooth, we know that $S$ is $C^\infty$ smooth. If we combine Lemma C.4, Lemma C.7 and Proposition H.4, then we can directly show the following lemma:

**Lemma C.8.** *Define $\mathcal{X}_1 = \phi_1(S)$. For $K_0^{\mathrm{NTK}}$ and $K_0^{\mathrm{RFK}}$ defined on $S$, we have the following equivalence:*

$$
\begin{aligned}
\mathcal{H}_0^{\mathrm{RF}}(S) &\cong H^{\frac{d+3}{2}}(\mathcal{X}_1) \circ \phi_1, \quad \text{and} \\
\mathcal{H}_0^{\mathrm{NT}}(S) &\cong H^{\frac{d+1}{2}}(\mathcal{X}_1) \circ \phi_1.
\end{aligned}
\tag{58}
$$

Now we can obtain the interpolation relationship between $\mathcal{H}_0^{\mathrm{RF}}(S)$ and $\mathcal{H}_0^{\mathrm{NT}}(S)$.

**Lemma C.9.** *Suppose $s \geq 0$. We have*

$$
[\mathcal{H}_0^{\mathrm{NT}}(S)]^s \cong [\mathcal{H}_0^{\mathrm{RF}}(S)]^{\frac{s(d+1)}{d+3}}
$$

*Proof.* Define $\mathcal{X}_1 = \phi_1(S)$. Let $\sigma$ be the uniform measure on $\mathbb{S}^d$. Recalling (51), we have the interpolation on $\mathcal{X}_1$ with lebesgue measure denoted by $\mu_1$:

$$
[H^{\frac{d+3}{2}}(\mathcal{X}_1)]^{\frac{s(d+1)}{d+3}} \cong H^{\frac{s(d+1)}{2}}(\mathcal{X}_1) \cong [H^{\frac{d+1}{2}}(\mathcal{X}_1)]^s
\tag{59}
$$

that is

$$
\left( L^2(\mathcal{X}_1, \mu_1), H^{\frac{d+3}{2}}(\mathcal{X}_1) \right)_{\frac{s(d+1)}{d+3}, 2} \cong \left( L^2(\mathcal{X}_1, \mu_1), H^{\frac{d+1}{2}}(\mathcal{X}_1) \right)_{s, 2}
\tag{60}
$$

Since $f \mapsto f \circ \phi_1$ is an isometric isomorphism, we have

$$
\left( L^2(\mathcal{X}_1, \mu_1) \circ \phi_1, H^{\frac{d+3}{2}}(\mathcal{X}_1) \circ \phi_1 \right)_{\frac{s(d+1)}{d+3}, 2} \cong \left( L^2(\mathcal{X}_1, \mu_1) \circ \phi_1, H^{\frac{d+1}{2}}(\mathcal{X}_1) \circ \phi_1 \right)_{s, 2}
\tag{61}
$$

Recall that $\mathcal{X}$ is bounded and thus $\mathcal{X}_1 = \phi_1(\phi(\mathcal{X}))$ is bounded. Therefore, the Jacobian $J\phi_1^{-1}$ satisfies $c \leq |J\phi_1^{-1}| \leq C$ for some constant $c$ and $C$. It is easy to verify that $L^2(\mathcal{X}_1, \mu_1) \circ \phi_1 = L^2(S, \mu_1 \circ \phi_1) \cong L^2(S, \sigma)$. Finally, with Lemma C.8, Lemma H.5 and Lemma H.6, we have

$$
[\mathcal{H}_0^{\mathrm{RF}}(S)]^{\frac{s(d+1)}{d+3}} \cong [\mathcal{H}_0^{\mathrm{NT}}(S)]^s
\tag{62}
$$

with respect to the uniform measure $\sigma$ on $S$. $\qquad\square$

## C.5 Smoothness of Gaussian process

Lemma C.9 provides the interpolation relationship between $\mathcal{H}_0^{\mathrm{NT}}(S)$ and $\mathcal{H}_0^{\mathrm{RF}}(S)$. By the kernel transformation relationship of NTK and RFK from $\mathbb{R}^d$ and to $\mathbb{S}^d$ as described in (42), we can also derive the interpolation relationship of $\mathcal{H}^{\mathrm{NT}}$ and $\mathcal{H}^{\mathrm{RF}}$. It will help for us to derive the smoothness of $f^{\mathrm{GP}}$.

**Lemma C.10** (Interpolation of RKHSs). *Suppose $s > 0$. We have*

$$
[\mathcal{H}^{\mathrm{NT}}(\mathcal{X})]^s \cong [\mathcal{H}^{\mathrm{RF}}(\mathcal{X})]^{\frac{s(d+1)}{d+3}}
$$

*with respect to measure $\mu$ on $\mathcal{X}$ which has Lebesgue density $c \leq p(x) \leq C$.*

*Proof.* Define a function $\rho(x) = \|\widetilde{x}\|$ on $\mathcal{X}$. Define measure $\nu$ on $\mathcal{X}$ such that the Radon-Nikodym derivative satisfies $\frac{\mathrm{d}\nu}{\mathrm{d}\mu} = \rho^2$. We consider measure $\nu \circ \phi$ on $S$ as well as measure $\mu$ on $\mathcal{X}$, and then define a map $I : [\mathcal{H}_0^{\mathrm{NT}}(S)]^s \to [\mathcal{H}^{\mathrm{NT}}(\mathcal{X})]^s$:

$$
I : f \mapsto \rho \cdot (f \circ \phi).
\tag{63}
$$

Now we prove $I$ is an isometric isomorphism. We first show that for any eigen pair $(f, \lambda)$ of $(K_0^{\mathrm{NTK}}, S, \nu \circ \phi)$, $(If, \lambda)$ is also an eigen pair of $(K^{\mathrm{NTK}}, \mathcal{X}, \mu)$. Actually, for eigen pair $(f, \lambda)$ we have

$$
\int_S K_0^{\mathrm{NTK}}(x, y) f(y) \, \mathrm{d}(\nu \circ \phi)(y) = \lambda f(z).
\tag{64}
$$

We perform a transformation of the integral domain,

$$
\int_{\mathcal{X}} K_0^{\mathrm{NTK}}(\phi(x), \phi(y)) f(\phi(y)) \mathrm{d}\nu(y) = \lambda f(\phi(x))
$$
$$
= \int_{\mathcal{X}} K_0^{\mathrm{NTK}}(\phi(x), \phi(y)) f(\phi(y)) \rho^2(y) \mathrm{d}\mu(y)
$$

(65)

Recalling the transformation between $K_0^{\mathrm{NTK}}$ and $K^{\mathrm{NTK}}$ in (42), we have

$$
\int_{\mathcal{X}} \rho(x) K_0^{\mathrm{NTK}}(\phi(x), \phi(y)) f(\phi(y)) \rho^2(y) \mathrm{d}\mu(y) = \lambda \rho(x) f(\phi(x))
$$
$$
= \int_{\mathcal{X}} K(x, y) f(\phi(y)) \rho(y) \mathrm{d}\mu(y)
$$

(66)

These transformations are both reversible. Therefore, through the structure of real interpolation space as described in (6), we can see $I$ is an isometric isomorphism . In the same way, there exist isometric isomorphism $I' : [\mathcal{H}_0^{\mathrm{RF}}(S)]^{\frac{s(d+1)}{d+3}} \to [\mathcal{H}^{\mathrm{RF}}(\mathcal{X})]^{\frac{s(d+1)}{d+3}}$:

$$
I' : f \mapsto \rho \cdot (f \circ \phi).
$$

(67)

Combined the result in Lemma C.9, the Lemma is proved. $\qquad\square$

Now we are ready to give the smoothness of Gaussian process $f^{\mathrm{GP}}$. We remind the reader that $\mathcal{H}^{\mathrm{NT}}$ and $\mathcal{H}^{\mathrm{RF}}$ are abbreviations used for denoting $\mathcal{H}^{\mathrm{NT}}(\mathcal{X})$ and $\mathcal{H}^{\mathrm{RF}}(\mathcal{X})$, respectively.

*Proof of Theorem 4.2.* Let $t = \frac{s(d+1)}{d+3}$ to simplify the notation. By Lemma C.10, we have

$$
[\mathcal{H}^{\mathrm{NT}}]^s \cong [\mathcal{H}^{\mathrm{RF}}]^t.
$$

(68)

Recalling the structure of interpolation space, we suppose $[\mathcal{H}^{\mathrm{RF}}]^t$ can be written as

$$
[\mathcal{H}^{\mathrm{RF}}]^t = \left\{ \sum_{i \in \mathbb{N}} c_i \lambda_i^{\frac{t}{2}} e_i \, \middle| \, \sum_{i \in \mathbb{N}} c_i^2 < \infty \right\}.
$$

(69)

Recall that $f^{\mathrm{GP}}$ represents a random function defined on $(\Omega, \mathcal{F}, \mathbf{P})$, where each $\omega \in \Omega$ corresponds to a path function $f_\omega^{\mathrm{GP}} : \mathcal{X} \to \mathbb{R}$. We can express this in the orthonormal basis as $f_\omega^{\mathrm{GP}} = \sum_{i \in \mathbb{N}} a_i(\omega) \lambda_i^{\frac{t}{2}} e_i$, where

$$
a_i(\omega) = \langle f_\omega^{\mathrm{GP}}, \lambda_i^{\frac{t}{2}} e_i \rangle_{[\mathcal{H}^{\mathrm{RF}}]^t} = \lambda_i^{-\frac{t}{2}} \int f_\omega^{\mathrm{GP}} e_i(x) \mathrm{d}\mu(x).
$$

Recall that as defined in Lemma 3.2, $f^{\mathrm{GP}}$ has the distribution $\mathcal{GP}(0, K^{\mathrm{RFK}})$. From this, we can acquire the joined distribution for $a_i$. Firstly, let us compute the covariance:

$$
\mathrm{Cov}(a_i, a_j) = \mathbf{E}[a_i, a_j]
$$
$$
= \mathbf{E}\left[ \lambda_i^{-t/2} \lambda_j^{-t/2} \int_{\mathcal{X}} \int_{\mathcal{X}} f^{\mathrm{GP}}(x) f^{\mathrm{GP}}(y) e_i(x) e_i(y) \, \mathrm{d}\mu(x) \mathrm{d}\mu(y) \right]
$$
$$
= \lambda_i^{-t/2} \lambda_j^{-t/2} \int_{\mathcal{X}} \int_{\mathcal{X}} \mathbf{E}\left[ f^{\mathrm{GP}}(x) f^{\mathrm{GP}}(y) \right] e_i(x) e_i(y) \, \mathrm{d}\mu(x) \mathrm{d}\mu(y)
$$
$$
= \lambda_i^{-t/2} \lambda_j^{-t/2} \int_{\mathcal{X}} \int_{\mathcal{X}} K^{\mathrm{RFK}}(x, y) e_i(x) e_i(y) \, \mathrm{d}\mu(x) \mathrm{d}\mu(y)
$$
$$
= \lambda_i^{-(1-t)/2} \lambda_j^{-(1-t)/2} \mathbf{1}_{\{i=j\}}.
$$

(70)

The exchange of integration is accomplished by Fubini's theorem since $K^{\mathrm{RFK}}$ is a bounded kernel function, and both $e_i$ and $e_j$ are $L_2$ integrable. Moreover, as $f^{\mathrm{GP}}$ is a Gaussian process, we finally get $a_i \sim N(0, \lambda_i^{1-t})$ for $i \in \mathbb{N}$, and $a_i, a_j$ are independent for any $i \neq j$. Consequently, we can directly derive that

$$
\left\| f^{\mathrm{GP}} \right\|_{[\mathcal{H}_{\mathrm{NT}}]^s}^2 = \sum_{i \in \mathbb{N}} \lambda_i^{1-t} Z_i^2,
$$

(71)

where $\{Z_i\}$ indicates a collection of independent and identically distributed standard Gaussian random variables. Finally, as Lemma C.3 establishes the eigenvalue decay rate as

$$
\lambda_i \asymp i^{\frac{d+3}{d}},
$$

(72)

it is direct to prove the theorem.

**Part 1.** When $s < \frac{3}{d+1}$, we have $\frac{d+3}{d} \cdot (1-t) > 1$ and thus

$$\mathbf{E} \left\| f^{\mathrm{GP}} \right\|_{[\mathcal{H}_{\mathrm{NT}}]^s}^2 \asymp \sum_{i \in \mathbb{N}} i^{-\frac{d+3}{d} \cdot (1-t)} < +\infty. \tag{73}$$

Consequently we have $\mathbf{P} \left( \left\| f^{\mathrm{GP}} \right\|_{[\mathcal{H}_{\mathrm{NT}}]^s}^2 < \infty \right) = 1$.

**Part 2.** When $s \geq \frac{3}{d+1}$, we ascertain that $\frac{d+3}{d} \cdot (1-t) \leq 1$ and consequently

$$\mathbf{E} \left\| f^{\mathrm{GP}} \right\|_{[\mathcal{H}_{\mathrm{NT}}]^s}^2 \asymp \sum_{i \in \mathbb{N}} i^{-\frac{d+3}{d} \cdot (1-t)} = +\infty. \tag{74}$$

Denote by $X_n = \sum_{i=1}^n \lambda_i^{1-t} Z_i^2$. We then obtain

$$\mathbf{E} X_n = \sum_{i=1}^n \lambda_i^{1-t}, \quad \mathrm{Var} X_n = \sum_{i=1}^n 2\lambda_i^{1-t}. \tag{75}$$

We can thus derive that

$$\mathbf{P}(X_n \leq \frac{\mathbf{E} X_n}{2}) \leq \mathbf{P}(|X_n - \mathbf{E} X_n| \geq \frac{\mathbf{E} X_n}{2}) \leq \frac{4 \mathrm{Var} X_n}{[\mathbf{E} X_n]^2} = \frac{8}{\sum_{i=1}^n \lambda_i^{1-t}}. \tag{76}$$

Given that $\left\| f^{\mathrm{GP}} \right\|_{[\mathcal{H}_{\mathrm{NT}}]^s}^2 \geq X_n$ for any $n \in \mathbb{N}_+$, we have

$$\mathbf{P}(\left\| f^{\mathrm{GP}} \right\|_{[\mathcal{H}_{\mathrm{NT}}]^s}^2 = \infty) = \lim_{M \to \infty} \mathbf{P}(\left\| f^{\mathrm{GP}} \right\|_{[\mathcal{H}_{\mathrm{NT}}]^s}^2 \geq M) \geq 1 - \lim_{n \to \infty} \mathbf{P}(X_n \leq \frac{\mathbf{E} X_n}{2}) = 1. \tag{77}$$

This completes the proof.

$\square$

## D   Proof of Theorem 4.3

With the findings from Theorem 4.2, Proposition 4.1, and Proposition B.4, the influence of non-zero initialization could be interpreted in terms of a misspecified spectral algorithms problem. To apply Proposition 2.2, it only remains to determine the embedding index of $\mathcal{H}^{\mathrm{NT}}$. Now, let's proceed to do so.

### D.1   Embedding index of $\mathcal{H}^{\mathrm{NT}}$

Recall that the Proposition 2.2 requires the embedding index of $\mathcal{H}^{\mathrm{NT}}$ on $\mathcal{X}$ under the probability measure $\mu$. Fortunately, the embedding index of the Sobolev space has been previously established by Zhang et al. [55], which is helpful to simplify our proof.

**Lemma D.1** (Zhang et al. [55] Section 4.2, Embedding index of Sobolev space). *Suppose $r > \frac{d}{2}$. For a bounded open set $\mathcal{X} \subset \mathbb{R}^d$ and Lebesgue measure $\mu$, the embedding index of $H^r(\mathcal{X})$ equals $\frac{d}{2r}$.*

Since we have established the relationship between $\mathcal{H}^{\mathrm{NT}}$ and the Sobolev space, we can easily get the embedding index through a similar way used in the proof of Lemma C.10.

**Lemma D.2** (Embedding index of NTK). *Suppose that the density function $p(x)$ of probability measure $\mu$ satisfies the condition $c \leq p(x) \leq C$, where $c$ and $C$ are positive constants. The embedding index of $\mathcal{H}^{\mathrm{NT}}(\mathcal{X})$ with respect to $\mu$ is concluded to be $\frac{d}{d+1}$.*

We omit this proof as it can be carried out in the same manner as Lemma C.10. Here, we provide only the structure. First, the embedding index of $H^{\frac{d+1}{2}}(S)$ is $\frac{d}{d+1}$ ( Lemma D.1 and Lemma C.7). Second, the embedding index of $\mathcal{H}_0^{\mathrm{NT}}(S)$ is $\frac{d}{d+1}$ (Lemma C.4). Third, the embedding index of $\mathcal{H}^{\mathrm{NT}}(\mathcal{X})$ is $\frac{d}{d+1}$ since $I : f \mapsto \rho \cdot (f \circ \phi)$ is isometric isomorphism both from $\mathcal{H}_0^{\mathrm{NT}}(S)$ to $\mathcal{H}^{\mathrm{NT}}(\mathcal{X})$ and from $L^\infty(S, \nu' \circ \phi)$ to $L^\infty(\mathcal{X}, \mu)$, where measure $\nu'$ is defined on $\frac{d\nu'}{d\mu} = \rho$ (an argument similar to that in the proof of Lemma C.10).

## D.2 Proof of Theorem 4.3

*Proof of Theorem 4.3.* Recall that Proposition 4.1 elucidates the impact of non-zero initialization. Namely, the generalization error of the kernel gradient flow with an initialization of $f_0$ and a regression function $f^*$, is consequently equivalent to that of kernel gradient flow with initialization at $0$ and a regression function of $f^* - f_0$. On the other hand, Proposition B.4 demonstrated the uniform convergence from the network function to the kernel gradient flow predictor as the network width $m$ tends to infinity. Lemma D.2 verify the embbeding index condition in Proposition 2.2. Thus, we only need to verify the source condition that $f^{\mathrm{GP}} - f^*$ fulfills and to incorporate it with Proposition 2.2 in order to derive the generalization error of the kernel gradient flow.

Now we start the proof. Since the proofs for the cases $s \geq \frac{3}{d+1}$ and $0 < s < \frac{3}{d+1}$ are exactly the same, we will only provide the proof for the former case here. Through Theorem 4.2, we know for any $0 < r < \frac{3}{d+1}$, it follows that $\mathbf{E}\big\|f^{\mathrm{GP}}\big\|^2_{[\mathcal{H}^{\mathrm{NT}}]^r} = \sum \lambda_i^{1-r} < \infty$. Let $C_t = \mathbf{E}\big\|f^{\mathrm{GP}}\big\|_{[\mathcal{H}^{\mathrm{NT}}]^r}$. By the Markov inequality, for any $\delta' \in (0,1)$, we have with probability exceeding $1 - \delta'$, that

$$\big\|f^{\mathrm{GP}} - f^*\big\|_{[\mathcal{H}^{\mathrm{NT}}]^r} \leq \frac{R + C_r}{\delta'}. \tag{78}$$

Recall that the eigenvalue decay rate for $K^{\mathrm{NTK}}$ is $\frac{d+1}{d}$ as mentioned in Lemma C.2. Therefore, we have for any $\delta \in (0,1)$ and any $\varepsilon \in (0, \frac{3}{d+3})$, there exists $r < \frac{3}{d+1}$ such that $\frac{r\beta}{r\beta+1} = \frac{3}{d+3} - \varepsilon$ (i.e., $r = \frac{d^2 - 6d - 3d(d+3)\varepsilon}{3(d+1) + \varepsilon(d+1)(d+3)}$). Denote by $\widetilde{f}^* = f^* - f^{\mathrm{GP}}$ and $\widetilde{f}_t^{\mathrm{NTK}}$ be the kernel gradient flow predictor starts from initial value $0$. Through Proposition 2.2, We thus have

$$\Big\|\widetilde{f}_t^{\mathrm{NTK}} - \widetilde{f}^*\Big\|^2_{L^2} \leq \left(\frac{1}{\delta'} \ln \frac{6}{\delta}\right)^2 (R + C_r)^2 C' n^{-\frac{3}{d+3}+\varepsilon}, \tag{79}$$

holds with probability at least $1 - 2\delta'$ when $t \asymp n^{\frac{\beta}{r\beta+1}}$. Through Proposition 4.1, also we have

$$\big\|f_t^{\mathrm{NTK}} - f^*\big\|^2_{L^2} \leq \left(\frac{1}{\delta'} \ln \frac{6}{\delta}\right)^2 (R + C_r)^2 C' n^{-\frac{3}{d+3}+\varepsilon}, \tag{80}$$

holds with probability at least $1 - 2\delta'$. Through uniform convergence in Proposition B.4, we have

$$\sup_{t \geq 0} \left| \big\|f_t^{\mathrm{NN}} - f^*\big\|_{L^2} - \big\|f_t^{\mathrm{NTK}} - f^*\big\|^2_{L^2} \right| \leq \left(\frac{1}{\delta'} \ln \frac{6}{\delta}\right)^2 (R + C_r)^2 C' n^{-\frac{3}{d+3}+\varepsilon}, \tag{81}$$

with probability at least $1 - \delta'$ when $m$ is large enough. Therefore, with appropriate choice of $\delta'$ and $C'$, we can finish the proof.

$\square$

# E Proof of Theorem 4.4

In this section, we establish the generalization error rate lower bound in our problem. We incorporate a result delineated in [40], which systematically studies the learning rate of kernel regression. Prior to this, we take some preparatory work.

We assume $k$ is a dot-product kernel on $\mathbb{S}^d$ with eigenvalue decay rate $\beta$, with respect to the uniform measure. We notate the corresponding RKHS as $\mathcal{H}_k$. Then, we can verify that $\mathcal{H}_k$ satisties to the definition of *regular RKHS*, as detailed in [40]. Subsequently, the main theorem in [40] can be applied under our proposed settings, since $K_0^{\mathrm{NTK}}$ is a dot-product kernel defined on $\mathbb{S}^d$. It engenders the following lemma.

**Lemma E.1** (Generalization error lower bound)**.** *Assume $k$ is a dot-product kernel defined on $\mathbb{S}^d$, we have the interpolation space of its RKHS as $[\mathcal{H}_k]^s = \Big\{ \sum_{i\in\mathbb{N}} a_i \lambda_i^{\frac{s}{2}} e_i \big| \sum_{i\in\mathbb{N}} a_i^2 < \infty \Big\}$ where $\{e_i\}_{i\in\mathbb{N}}$ is the orthonormal basis of $L_2(\mathbb{S}^d, \sigma)$ and $\sigma$ denotes the uniform measure. Decompose the regression function $f^*$ over the series of basis:*

$$f^* = \sum_{i\in\mathbb{N}} f_i e_i. \tag{82}$$

We assume that $f^* \in [\mathcal{H}]^t$ holds for any $t < s$ for a given $s > 0$. Also, we we assume that

$$\sum_{i:\lambda > \lambda_i} |f_i|^2 = \Omega\left(\lambda^s\right). \tag{83}$$

We also assume that the noise term satisties $\mathbf{E}[|\epsilon|^2|x] = \sigma^2$ holds for $x \in \mathbb{S}^d$, a.e. Then, we define the main bias term in generalization error by

$$\mathcal{R}^2(t; f^*) = \sum_{i \in \mathbb{N}} e^{-2t\lambda_i} \lambda_i f_i^2, \tag{84}$$

and define the variance term by

$$\mathcal{N}(t) = \sum_{i \in \mathbb{N}} [\lambda_i e^{-t\lambda_i}]^2. \tag{85}$$

Fix the given input vectors of samples $X$. Consider the kernel gradient flow process detailed in (8) and let it start from $0$. For any choice of $t = t(n) \to \infty$, we have

$$\mathbf{E}\left[\left\|f_t^{\mathrm{GF}} - f^*\right\|_{L^2}^2 \Big| X\right] = \Omega_{\mathbf{P}}\left(\mathcal{R}^2(t; f^*) + \frac{1}{n}\mathcal{N}(t)\right), \tag{86}$$

With the lemma above, now we are ready to prove Theorem 4.4.

*Proof of Theorem 4.4.* By Proposition 4.1, we know the initial output function introduce an implicit bias term to the regression function. And thus the original problem is same as to consider a standard kernel gradient flow problem start from initial output zero with regression function $\tilde{f}^* = f^* - f^{\mathrm{GP}}$. Recall that $\mu$ is the uniform measure. On sphere, the RKHSs of dot-product kernels $K_0^{\mathrm{NTK}}$ and $K_0^{\mathrm{RFK}}$ are equivalent to corresponding Sobolev spaces through Lemma C.4. More precisely, suppose that we have chosen an orthonormal basis $\{e_i\}_{i \in \mathbb{N}}$ consisting of spherical harmonic polynomials. Then we have

$$\begin{aligned}
[\mathcal{H}_0^{\mathrm{NT}}]^t &= \left\{\sum_{i \in \mathbb{N}} a_i \omega_i^{t/2} e_i \Big| \sum_{i \in \mathbb{N}} a_i^2 < \infty\right\}, \\
[\mathcal{H}_0^{\mathrm{RF}}]^t &= \left\{\sum_{i \in \mathbb{N}} a_i \lambda_i^{t/2} e_i \Big| \sum_{i \in \mathbb{N}} a_i^2 < \infty\right\}
\end{aligned} \tag{87}$$

for any $t \geq 0$. Through Lemma C.2, we have the eigenvalue decay rate:

$$\omega_i \asymp i^{-\frac{d+1}{d}} \quad \text{and} \quad \lambda_i \asymp i^{-\frac{d+3}{d}}. \tag{88}$$

We denote by $\beta_1 = \frac{d+1}{d}$ and $\beta_2 = \frac{d+3}{d}$. Similar to the proof of Theorem 4.2, we write the Kosambi–Karhunen–Loève expansion of $\tilde{f}^*$:

$$\tilde{f}^* = \sum_{i \in \mathbb{N}} \widetilde{f}_i e_i = \sum_{i \in \mathbb{N}} (b_i - a_i)\lambda_i^{1/2} e_i, \tag{89}$$

where

$$a_i \overset{\text{i.i.d.}}{\sim} N(0, 1), \quad \text{and} \quad f^* = \sum_{i \in \mathbb{N}} b_i \lambda_i^{1/2} e_i \in [\mathcal{H}_0^{\mathrm{NT}}]^s. \tag{90}$$

Here $a_i$ is a sequence of independent standard Gaussian variables, and $b_i$ represents a sequence derived from the decomposition of $f^*$. With such decomposition, we can verify that (83) holds with probability $1 - \delta'$ for any $\delta' \in (0, 1)$. Denote by $g(\lambda) = \sum_{i:\lambda > \omega_i} |\widetilde{f}_i|^2$. Firstly, we have

$$\mathbf{E}[g(\lambda)] = \mathbf{E}\left[\sum_{i:\lambda > \omega_i} |\widetilde{f}_i|^2\right] \asymp \mathbf{E}\left[\sum_{i > \lfloor \lambda^{-\frac{d}{d+1}} \rfloor} |\widetilde{f}_i|^2\right] \gtrsim \lambda^{\frac{3}{d+1}}. \tag{91}$$

We also have the variance

$$\mathrm{Var}\left[g(\lambda)\right] = \mathrm{Var}\left[\sum_{i:\lambda > \omega_i} |\widetilde{f}_i|^2\right] \asymp \sum_{i:\lambda > \omega_i} \lambda_i + \sum_{i:\lambda > \omega_i} \lambda_i b_i^2 \lesssim \lambda^{\frac{3}{d+1}}. \tag{92}$$

Where the second term is controlled by the source condition assumption on $f^*$. Therefore, we have

$$\mathbf{P}\left(|g(\lambda) - \mathbf{E}[g(\lambda)]| \geq \mathbf{E}\left[\frac{g(\lambda)}{2}\right]\right) \leq \frac{4\mathrm{Var}[g(\lambda)]}{(\mathbf{E}g[(\lambda)])^2} = O(\lambda^{\frac{3}{d+1}}). \tag{93}$$

Define event $A(\lambda) = \{|g(\lambda) - \mathbf{E}[g(\lambda)]| \leq \mathbf{E}[g(\lambda)]/2\}$. For any $\delta' \in (0, 1)$, we choose a sequence $\widetilde{\lambda}_j$, such that $\widetilde{\lambda}_j = C'j^{-\frac{2(d+1)}{3}}$. Then we have

$$\mathbf{P}\left(\cup_{j \in \mathbb{N}} A\left(\widetilde{\lambda}_j\right)\right) \geq 1 - \sum_{j \in \mathbb{N}}[C']^{\frac{3}{d+1}}j^{-2}. \tag{94}$$

We can choose appropriate $C' > 0$ such that $\cup_{j \in \mathbb{N}} A(\widetilde{\lambda}_j)$ holds with probability at least $1 - \delta'$, we denote by event $A$. Conditioned on event $A$, for any $\widetilde{\lambda}_{j+1} \leq \lambda \leq \widetilde{\lambda}_j$, we have

$$g(\lambda) \geq g(\widetilde{\lambda}_{j+1}) \gtrsim \frac{1}{2}(\widetilde{\lambda}_{j+1})^{\frac{3}{d+1}} \quad \text{and} \quad \frac{\widetilde{\lambda}_{j+1}}{\widetilde{\lambda}_j} = \left(\frac{j}{j+1}\right)^{\frac{2(d+1)}{3}} \tag{95}$$

which shows that

$$g(\lambda) \gtrsim \frac{1}{2}(\widetilde{\lambda}_{j+1})^{\frac{3}{d+1}} \gtrsim \frac{1}{2}\left[\frac{j}{j+1}\right]^2 \lambda^{\frac{3}{d+1}}. \tag{96}$$

Therefore, we finish the proof of (83).

Then, we turns to the calculation of generalization error lower bound. First, we plug in the decomposition and calculate the bias term $\mathcal{R}(t; \tilde{f}^*)$:

$$\mathcal{R}^2(t; \tilde{f}^*) = \sum_{i \in \mathbb{N}} e^{-2t\lambda_i}\lambda_i(a_i^2 - 2b_i a_i + b_i^2). \tag{97}$$

Recalling that the eigenvalue decay rate is denoted by $\beta$, it follows that

$$\mathbf{E}\left[\mathcal{R}^2(t; \tilde{f}^*)\right] \geq \sum_{i \in \mathbb{N}} e^{-2t\lambda_i}\lambda_i \asymp \sum_{i \in \mathbb{N}} e^{-2ti^{-\beta_1}}i^{-\beta_2} \asymp t^{\frac{1}{\beta_1} - \frac{\beta_2}{\beta_1}}. \tag{98}$$

Also, the variance of $\mathcal{R}^2(t; \tilde{f}^*)$ follows that

$$\mathrm{Var}(\mathcal{R}^2(t; \tilde{f}^*)) \lesssim \sum_{i \in \mathbb{N}} e^{-4t\lambda_i}\lambda_i^2 + \sum_{i \in \mathbb{N}} e^{-4t\lambda_i}b_i^2\lambda_i^2, \tag{99}$$

Here we introduce the denotations:

$$V_0 := \sum_{i \in \mathbb{N}} e^{-4t\lambda_i}2\lambda_i^2, \quad \text{and} \quad V_2 := \sum_{i \in \mathbb{N}} e^{-4t\lambda_i}b_i^2\lambda_i^2. \tag{100}$$

We then have

$$V_0 = \sum_{i \in \mathbb{N}} e^{-4t\lambda_i}\lambda_i^2 \asymp \sum_{i \in \mathbb{N}} e^{-4ti^{-\beta_1}}i^{-2\beta_2} \asymp t^{\frac{1}{\beta_1} - 2\frac{\beta_2}{\beta_1}}. \tag{101}$$

As to $V_2$, we first recall that the smoothness of $f^*$ lead to the following inequality:

$$\sum_{i \in \mathbb{N}} b_i^2 i^{-1} < \infty, \tag{102}$$

which implies that

$$\sum_{i \in \mathbb{N}} b_i^4 i^{-2} < \infty. \tag{103}$$

Now we turn to the evaluation of $V_2$:

$$\begin{aligned}
V_2 &= \sum_{i \in \mathbb{N}} e^{-4t\lambda_i}b_i^2\lambda_i^2 \asymp \sum_{i \in \mathbb{N}} e^{-4ti^{-\beta_1}}b_i^2 i^{-2\beta_2} = \sum_{i \in \mathbb{N}} e^{-4ti^{-\beta_1}}b_i^2 i^{-1} i^{-2\beta_2+1} \\
&\leq \sqrt{\sum_{i \in \mathbb{N}} e^{-8ti^{-2\beta_1}}i^{-4\beta_2+2} \sum_{i \in \mathbb{N}} b_i^4 i^{-2}} \lesssim t^{\frac{1}{2\beta_1} - 2\frac{\beta_2}{\beta_1} + \frac{1}{\beta_1}}.
\end{aligned} \tag{104}$$

It is worth noting that we use Cauchy's inequality to derive the upper bound above. With the control of $V_0$ and $V_2$, we have

$$\text{Var}(\mathcal{R}^2(t; \tilde{f}^*)) \asymp V_0 + V_2 \lesssim t^{\frac{3}{2\beta_1} - 2\frac{\beta_2}{\beta_1}} \tag{105}$$

Consequently, by Chebyshev's inequality, we directly have

$$\mathbf{P}\left(|\mathcal{R}^2(t; \tilde{f}^*) - \mathbf{E}\left[\mathcal{R}^2(t; \tilde{f}^*)\right]| \geq \mathbf{E}\left[\mathcal{R}^2(\tilde{f}^*)\right]/2\right) \leq \frac{4\text{Var}(\mathcal{R}^2(t; \tilde{f}^*))}{\left(\mathbf{E}\left[\mathcal{R}^2(t; \tilde{f}^*)\right]\right)^2} = O\left(t^{-\frac{1}{2\beta_1}}\right). \tag{106}$$

Since $t = t(n) \to +\infty$, we have

$$\mathcal{R}^2(t; \tilde{f}^*) = \Omega_{\mathbf{P}}\left(t^{\frac{1}{\beta_1} - \frac{\beta_2}{\beta_1}}\right). \tag{107}$$

In the same way, we also have the bound of variance term $\mathcal{N}(t)$.

$$\mathcal{N}(t) \asymp \frac{1}{n} t^{\frac{1}{\beta_1}}. \tag{108}$$

Finally, apply Lemma E.1 and Proposition 3.3. We derive that for any $\delta > 0$, as long as $n$ is large enough and $m$ is large enough, for any choice of $t = t(n) \to \infty$, with probability at least $1 - \delta$ we have

$$\mathbf{E}\left[\|f_t^{\text{NN}} - f^*\|_{L^2}^2 | X\right] = \Omega\left(\mathcal{R}^2 + \frac{1}{n}\mathcal{N}\right) = \Omega\left(t^{\frac{1}{\beta_1} - \frac{\beta_2}{\beta_1}} + \frac{1}{n} t^{\frac{1}{\beta_1}}\right) = \Omega\left(n^{-\frac{3}{d+3}}\right). \tag{109}$$

Thus the theorem is proved. $\qquad\square$

**Remark E.2.** In Proposition 3.3, we consider the situation that both input $X$ and output $Y$ of samples are fixed, while in the proof above we require that only $X$ is fixed. However, the conclusion of Proposition 3.3 still holds when $Y$ of samples is random. This is because the noise term has a finite second moment. Also, the change of domain from $\mathcal{X}$ to $\mathbb{S}^d$ will not affect the uniform convergence result.

## F  Details in artificial data experiments

Fixing the dimension of data as $d = 5, 10$. We draw samples for variable $x$ from the standard Gaussian distribution $\mathcal{N}(0, I_d)$, which are consequently standardized to lie on the surface of the unit hypersphere $\mathbb{S}^d$. The dependent variable $y$ is formulated as:

$$y = f(x) + \varepsilon, \tag{110}$$

where $f(x) = \left(\sum_{j=1}^d x_j\right)^2$, $\varepsilon \sim \mathcal{N}(0, \sigma^2)$ and $\sigma = 0.2$. The function $f$ exhibits notable smoothness, since it can be linearly represented in terms of the first few spherical harmonic polynomials on $\mathbb{S}^d$ [9] and the fact that $K_0^{\text{NTK}}$ is a dot-product kernel. We consider fully-connected network with one singular hidden layer, choosing $m = 20 * n$ to ensure large enough width. Consistent to previous sections, we choose ReLU function as the non-linear activation and train the network using Gradient Descent for a sufficiently long time. We record the generalization error at each moment and define the moment of minimum generalization error as the final generalization error. This is done to align with the early stopping strategy mentioned in the Theorem 4.3.

## G  Details in real data experiments

In this subsection, we will provide the theoretical basis of the method which approximates the smoothness of the goal function of real dataset. Let $\mathcal{X} \subset \mathbb{R}^d$ be a bounded domain. Given a reproduce kernel $k(\cdot, \cdot)$ on $\mathcal{X}$ and a probability measure $\mu$. Denote the RKHS by $\mathcal{H}_k = \left\{\sum_{i \in \mathbb{N}} a_i \lambda_i^{\frac{1}{2}} e_i \middle| \sum_{i \in \mathbb{N}} a_i^2 < \infty\right\}$. We assume that there is a function $f : \mathcal{X} \to \mathbb{R}$ and a probability density $\mu$ on $\mathcal{X}$. Suppose that the samples satisfies $y = f(x)$, then $f$ has the decomposition:

$$f = \sum_{i \in \mathbb{N}} \theta_i e_i. \tag{111}$$

The smoothness $\alpha_f = \alpha(f,k)$ depends on the coefficients $c_i$: if we have $\theta_i \asymp i^{-d_c}$ and $\lambda_i \asymp i^{-d_\lambda}$, then we derive the smoothness: $\alpha_f = \frac{2d_c - 1}{d_\lambda}$.

We consider $n$ samples $\{(x_i, y_i)\}_{i=1}^n$. The Gram matrix $k(X, X)$ can be decomposed as

$$k(X, X) = \phi \Sigma \phi^T, \quad \text{and} \quad \frac{1}{n} \phi \phi^T = I_n. \tag{112}$$

In this regard, we can utilize the eigenvalue of the empirical kernel matrix to estimate the eigenvalue of the kernel function, since previous work has shown the convergence of eigenvalue when $n$ is large enough [32]. Through the decomposition $Y = \phi c$ (i.e., $c = \frac{1}{n} \phi^T Y$) and the approximation $c_i \approx \theta_i$, we can roughly estimate the eigenvalue decay rate when $n$ is large enough with respect to $i$:

$$\sum_{k=i}^n c_k^2 \asymp i^{-\alpha_f d_\lambda}. \tag{113}$$

In our experiments, we let $n = 3000$. Namely, an arbitrary selection of 3000 samples was made from the each dataset. We did not use all samples in the datasets, because $n = 3000$ is already sufficient to calculate the decay rate of eigenvalues. We consider the NTK of a one-hidden-layer fully connected network as the kernel $k$. The results is shown in Figure 2, Figure 3 and Figure 4 for the three datasets, respectively. In each figure, the scatter plot shows the log value of the summed squares of each $c_k$ (for $i \leq k \leq n$ as per equation (113)) against $\log_{10} i$ on the x-axis. Also, the dashed line represents the corresponding least-square regression fitting using index $i$ smaller than 2700. Theoretically, the slope of the dashed line will be $-\alpha_f d_\lambda$.

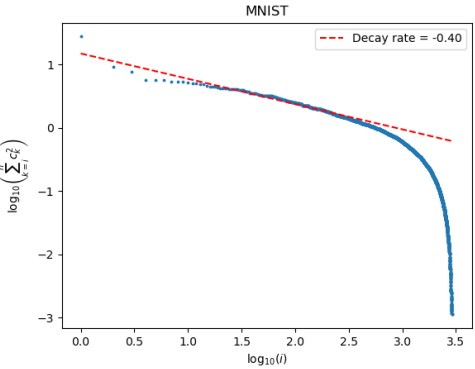

Figure 2: Decay curve of the logarithm of sum of squared coefficients for NMIST.

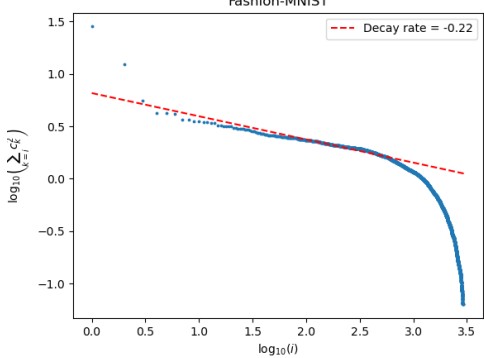

Figure 3: Decay curve of the logarithm of sum of squared coefficients for Fashion-NMIST.

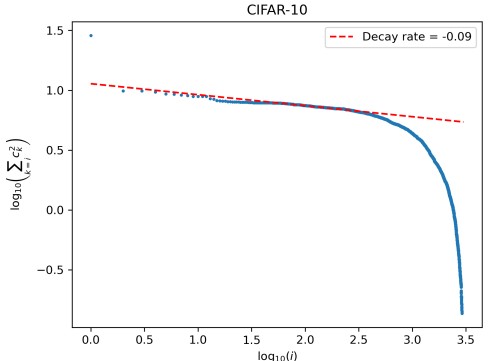

Figure 4: Decay curve of the logarithm of sum of squared coefficients for CIFAR-10.

# H    Technical Lemmas

In this section, we introduce a series of technical lemmas. These will be helpful in our proof, and many of the lemmas have been established by prior researchers.

**Lemma H.1** (Change of measure [41] ). *For a positive definite kernel $k$ defined on a compact set $\mathcal{X}$, it has the same eigenvalue decay rate under two measure $\nu$ and $\sigma$:*

$$\lambda_i(K_0^{\mathrm{NTK}}, \mathcal{X}, \nu) \asymp \lambda_i(K_0^{\mathrm{NTK}}, \mathcal{X}, \sigma)$$

*if the Radon derivative $p = \frac{\mathrm{d}\nu}{\mathrm{d}\sigma}$ exists and $c \le p \le C$ holds for some positive constant $c$ and $C$.*

**Lemma H.2** (Skorohod's Representation Theorem). *Suppose that a sequence of probability distribution $\{F_n\}$ converges weakly to $F$ and $F$ has a separable support. Then there exist random variables $X_n$ and $X$, defined on a new probability space $(\Omega', \mathcal{F}, \mathbf{P})$, such that the distribution of $X_n$ is $F_n$, the distribution of $X$ is $P$, and $X_n \to X$ holds almost surely.*

**Lemma H.3** (Restriction of RKHS [3]). *Suppose $\mathcal{H}_k$ is a RKHS defined on $E$ with the norm $\|\cdot\|_{\mathcal{H}_k}$, then $k|_\Omega$ restricted to a subset $\Omega \subset E$ is the reproducing kernel of space $\{f' = f|_\Omega|, f \in \mathcal{H}_k\}$ with norm defined by*

$$\|f'\| = \min_{f|_\Omega = f'} \|f\|_{\mathcal{H}_k}. \tag{114}$$

The following proposition is a direct proposition of Lemma H.3:

**Proposition H.4** (Equivalence of RKHS under resriction). *Assume RKHSs $\mathcal{H}_1 \cong \mathcal{H}_2$ are defined on $E$. Write $\Omega$ as a subset of $E$. Then we have $\mathcal{H}_1|_\Omega \cong \mathcal{H}_2|_\Omega$.*

The following two lemmas are common in real interpolation:

**Lemma H.5** (Equivalence of interpolation spaces). *Suppose $0 < s < 1$. Denote $L^2 = L^2(\mathcal{X}, \mu)$ for abbreviation. If we have RKHSs $\mathcal{H}_1 \cong \mathcal{H}_2$, then $(L^2, \mathcal{H}_1)_{s,2} \cong (L^2, \mathcal{H}_2)_{s,2}$.*

*Proof.* To prove the lemma, we only need to prove that the embedding $(\mathcal{H}_1, L^2)_{s,2} \hookrightarrow (L^2, \mathcal{H}_2)_{s,2}$ and $(L^2, \mathcal{H}_2)_{s,2} \hookrightarrow (L^2, \mathcal{H}_2)_{s,2}$ are both bounded.

First we prove that $\left\|(L^2, \mathcal{H}_1)_{s,2} \hookrightarrow (L^2, \mathcal{H}_2)_{s,2}\right\| \le C_1$ where $C_1$ is an absolute positive constant.

For any $x \in L^2 + \mathcal{H}_1$, define the K-functional

$$K_1(t; x) = \inf_{x_0 + x_1 = x; x_0 \in L^2, x_1 \in \mathcal{H}_1} (\|x_0\|_{L^2} + t\|x_1\|_{\mathcal{H}_1});$$

$$K_2(t; x) = \inf_{x_0 + x_1 = x; x_0 \in L^2, x_1 \in \mathcal{H}_2} (\|x_0\|_{L^2} + t\|x_1\|_{\mathcal{H}_2}).$$

Since $\mathcal{H}_1 \cong \mathcal{H}_2$, for any $x \in \mathcal{H}_1$, we have $\|x\|_{\mathcal{H}_2} \le C\|x\|_{\mathcal{H}_1}$. Thus we have

$$K_2(t; x) \le \inf_{x_0 + x_1 = x; x_0 \in L^2, x_1 \in \mathcal{H}_1} (\|x_0\|_{L^2} + Ct\|x_1\|_{\mathcal{H}_1}) = K_1(Ct; x);$$

Then we have

$$\begin{aligned}
\|x\|_{(L^2, \mathcal{H}_2)_{s,2}} &= \int_0^\infty [t^{-s} K_2(t; x)]^2 \, \frac{dt}{t} \\
&\leq \int_0^\infty [t^{-s} K_1(Ct; x)]^2 \, \frac{dt}{t} \\
&\leq C^{2s} \int_0^\infty [(Ct)^s K_1(Ct; x)]^2 \, \frac{d(Ct)}{Ct} \\
&= C^{2s} \|x\|_{(L^2, \mathcal{H}_1)_{s,2}}
\end{aligned} \tag{115}$$

Let $C_1 = C^{2s}$, we have the canonical injection satisfies $\left\| (L^2, \mathcal{H}_1)_{s,2} \hookrightarrow (L^2, \mathcal{H}_2)_{s,2} \right\|_{\mathrm{op}} \leq C_1$. Also, since $\mathcal{H}_1 \cong \mathcal{H}_2$, for any $x \in \mathcal{H}_2$, we have $\|x\|_{\mathcal{H}_1} \leq c\|x\|_{\mathcal{H}_2}$. We can prove $\left\| (L^2, \mathcal{H}_2)_{s,2} \hookrightarrow (L^2, \mathcal{H}_1)_{s,2} \right\|_{\mathrm{op}} \leq C_2$ in the same way. Then, we finish the proof. $\qquad \square$

**Lemma H.6** (Equivalence of interpolation spaces). *Suppose $0 < s < 1$. Denote $\mathcal{H}$ be a RKHS and $\mu, \nu$ be measures on set $\mathcal{X}$. If we have $L^2(\mathcal{X}, \mu) \cong L^2(\mathcal{X}, \nu)$, then $(L^2(\mathcal{X}, \mu), \mathcal{H})_{s,2} \cong (L^2(\mathcal{X}, \nu), \mathcal{H})_{s,2}$.*

*Proof.* The proof in accomplished in the same way as Lemma H.5. $\qquad \square$

