# OpenReview forum: "On the Impacts of the Random Initialization in the Neural Tangent Kernel Theory"
_NeurIPS.cc/2024/Conference — NeurIPS 2024 poster_

### Official Review · Reviewer_Yc4R · 2024-07-11

**Soundness:** 3
**Presentation:** 2
**Contribution:** 3
**Rating:** 5
**Confidence:** 3

**Summary:**

In this paper, the authors discuss the impact of random initialization of deep neural networks in the neural tangent kernel (NTK) regime.
Precisely, the authors prove that in the case of standard random initialization, the network function in the finite wide limit, converges to the NTK predictor uniformly.
Then, by studying the behavior of the underlying Gaussian process, the authors establish upper and lower bounds for the generalization errors of DNNs in the NTK regime, in terms of the real interpolation space of the RKHS.

**Strengths:**

The paper studies an important problem.
The paper is in general well written and not difficult to follow.
The numerical results, at least those with artificial data, look compelling.

**Weaknesses:**

It is a bit difficult for me to evaluate this paper.
The technical results of Theorem 3.3 and 4.1 are not very surprising and their proofs also look more or less standard.
The result in Theorem 4.2 appear novel and interesting, but it is a bit difficult to position this in the existing literature of, kernel or DNN. I believe that the authors should make more efforts on that to make the message of this paper more clear.

**Questions:**

* Is Proposition 2.2 novel result? If yes, where is its proof?
* it appears that the notations of expectation are not aligned in Assumption 1 and line 178.
* The results of Theorem 3.3 and 4.1 are not very surprising and their proofs also look more or less standard
* The result in Theorem 4.2 looks interesting, but essentially focuses on the Gaussian process with zero mean and random feature kernel of covariance function. Also, is this the FIRST time such results are established (e.g., in terms of real interpolation space of the RKHS)? How should we compare this result to other kernels/methods/models? It may be helpful to compare this behavior to the literature.

---
I thank the authors for their detailed reply and clarification. This helps me have a better grasp of this presented results, I will increase my score accordingly.

I would like to ask the authors to carefully include the above discussions in a revised version of the paper, as also suggested by Reviewer 43yC.

**Limitations:**

I do not see any potential negative social impact of this work.

---

> ### Author Rebuttal · Authors · 2024-08-06
>
> Thank you for your careful reading and for raising these questions. We will summarize our main technical contribution and address the questions:
> ### **Summary of our technical contribution**
> *"The result in Theorem 4.2 appears novel and interesting, but it is a bit difficult to position this in the existing literature on kernel methods or deep neural networks. I believe that the authors should make more efforts to make the message of this paper clearer."*
>
> Thank you for your review and thoughtful comments. Your intuition is correct. Our paper discusses the impact of initialization on the generalization ability of network under NTK theory, with the most critical technical contribution being Theorem 4.2. This theorem addresses the smoothness of the Gaussian process approximated by the initial output network function with respect to the NTK. Now we discuss the challenges and contributions of Theorem 4.2:
> - **Challenge of Theorem 4.2**:
>   - If the input is uniformly distributed on the sphere $\mathbb{S}^d$ instead of $\mathcal{X}$, both NTK and RFK are inner-product kernels and their Mercer's decomposition shares the same eigen-function (i.e., spherical harmonic polynomials). In this situation, simple calculations on the eigenvalues would show that $f^{\mathrm{GP}}\in [\mathcal{H}^{\mathrm{NT}}]^{\frac{3}{d+1}}$.
>   - However, when the data is a general distribution supported in a bounded open set $\mathcal{X} \subset \mathbb{R}^d$. NTK and RFK will not possess the nice property anymore which makes the claim if $f^{\mathrm{GP}}\in [\mathcal{H}^{\mathrm{NT}}]^{\frac{3}{d+1}}$ unclear.
> - **Solution and Technical Contributions**:
>   - Before we clarify our solution and technical contributions, we first recollect some notations. For a sub-region $S\subset \mathbb{S}^d$, let $\mathcal{H}_0^{\mathrm{NT}}(\mathbb{S}^d)$ be the RKHS associated with the homogeneous NTK defined on $\mathbb{S}^d$ and $\mathcal{H}_0^{\mathrm{NT}}(S)=\mathcal{H}_0^{\mathrm{NT}}(\mathbb{S}^d) |_S$ be its restriction on the sub-region $S\subset \mathbb{S}^d$. We define $\mathcal{H}_0^{\mathrm{RF}}(\mathbb{S}^d)$ and $\mathcal{H}_0^{\mathrm{RF}}(S)$ similarly.
>   - We first show that, on a general $\mathcal{X}$, $\mathcal{H}^{\mathrm{RF}} \cong [\mathcal{H}^{\mathrm{NT}}]^\frac{d+3}{d+1}$ (where the $\cong$ is isomorphic as Hilbert space not RKHS).
>     - Let us fix a continuous and isomorphic mapping from $\mathcal{X} \subset \mathbb{R}^d$ to a sub-region $S \subset \mathbb{S}^d$. We then show the equivalence (as RKHS):
>       $$
>       \mathcal{H}^{\mathrm{\mathrm{RF}}}\cong \mathcal{H}_0^{\mathrm{\mathrm{RF}}}(S), \quad \mathcal{H}^{\mathrm{NT}}\cong \mathcal{H}_0^{\mathrm{NT}}(S).
>       $$
>       This equivalence is not as trivial as it appears.
>     - We then need to show that $[\mathcal{H}_0^{\mathrm{RF}}(S)]\cong [\mathcal{H}_0^{\mathrm{NT}}(S)]^{\frac{d+3}{d+1}}$.
>       Notice that the Mercer's decomposition of RFK and NTK share the same eigenfunctions (i.e., the spherical harmonic polynomials), we know that $[\mathcal{H}_0^{\mathrm{RF}}(\mathbb{S}^d)]\cong [\mathcal{H}_0^{\mathrm{NT}}(\mathbb{S}^d)]^{\frac{d+3}{d+1}}$. If the restriction $\mid_S$ and the interpolation $[]^{\frac{d+3}{d+1}}$ are commutative, then we are done. Unfortunately, this commutativity is not easy to verify in general.
>     - Fortunately, by a recent result [haas2023mind], we have $\mathcal{H}_0^{\mathrm{RF}}(\mathbb{S}^d) \cong W^{\frac{d+3}{2},2}(\mathbb{S}^d)$ and $\mathcal{H}_0^{\mathrm{NT}}(\mathbb{S}^d) \cong W^{\frac{d+1}{2},2}(\mathbb{S}^d)$ where $W^{s,2}$ denotes the usual fractional Sobolev space. Since Sobolev space is a more tractable object, we then carefully verify that $\mathcal{H}_0^{\mathrm{RF}}(S) = [\mathcal{H}_0^{\mathrm{NT}}(S)]^{\frac{d+3}{d+1}}$ and finish the proof. We would like to emphasize that there are several different definitions for the fractional Sobolev spaces and there are no similar results explicitly stated in the literature.
>   - By Lemma H.5, we have $[\mathcal{H}^{\mathrm{RF}}]^{\frac{3}{d+3}} \cong [[\mathcal{H}^{\mathrm{NT}}]^\frac{d+3}{d+1}]^{\frac{3}{d+3}} (\cong [\mathcal{H}^{\mathrm{NT}}]^{\frac{3}{d+1}})$.
>   - If we denote $\mathcal{H}^{\mathrm{RF}} = \left\lbrace \sum_{i \in \mathbb{N}} a_i \lambda_i^{\frac{1}{2}} e_i \mid \sum_{i \in \mathbb{N}} a_i^2 < \infty \right\rbrace $, then the K-L expansion gives us that $f^{\mathrm{GP}} = \sum_{i \in \mathbb{N}} Z_i \lambda_i^{\frac{1}{2}} e_i$ where $\lbrace Z_i\rbrace_{i \in \mathbb{N}}$ are i.i.d. standard Gaussian variables. We can verify that $\mathbf{P}( f^{\mathrm{GP}}\in [\mathcal{H}^{\mathrm{RF}}]^{t}) = 1$ for $t < \frac{3}{d+3}$ and that $\mathbf{P}( f^{\mathrm{GP}}\in [\mathcal{H}^{\mathrm{RF}}]^{t}) = 0$ for $t \geq \frac{3}{d+3}$. i.e., we have
>   $$
>   \mathbf{P}( f^{\mathrm{GP}}\in [\mathcal{H}^{\mathrm{NT}}]^{t}) = 1 \text{ for } t < \frac{3}{d+1} \text{ and } \mathbf{P}( f^{\mathrm{GP}}\in [\mathcal{H}^{\mathrm{NT}}]^{t}) = 0 \text{ for }t\geq \frac{3}{d+1}.
>   $$
> ### **Response to the Questions**:
>   1. **Proposition 2.2**: Proposition 2.2 is a direct corollary of Theorem 1 in [zhang2023optimality]. Theorem 1 in [zhang2023optimality] provides an upper bound on the generalization error for general spectral algorithms. We directly applied this to the kernel gradient flow, a special type of spectral algorithm.
>   2. **Consistency in Notation**: Thank you for your correction. We will use the expectation symbol $\mathbf{E}$ consistently.
>   3. **Theorem Designation**: Thank you for pointing out the issues with Theorem 3.3 and Theorem 4.1. These two theorems respectively provide conclusions on uniform convergence and the impact of the initial output function in kernel gradient flow. The proofs of them are standard and we will change their designation from Theorem to Proposition.
>   4. **Explanation of Theorem 4.2**: The explanation of the technical contribution of Theorem 4.2 is detailed in our response above. Thank you again for your valuable feedback.

---

> ### Author Response · Authors · 2024-08-07
> **Refernces of Rebuttal**
>
> (We apologize for the rebuttal reaching the 6000-word limit. We will provide the references below and appreciate your understanding for any inconvenience caused.)
>
> ### **References**
>
>
> [haas2023mind] Haas, M., et al. (2023). “Mind the spikes: Benign overfitting of kernels and neural networks in fixed dimension”.
>
>
>
> [zhang2023optimality] Zhang, H., Li, Y., and Lin, Q. (2023). “On the optimality of misspecified spectral algorithms”.

---

> ### Comment · Reviewer_Yc4R · 2024-08-09
>
> I thank the authors for their detailed reply and clarification.
> This helps me have a better grasp of this presented results, I will increase my score accordingly.
>
> I would like to ask the authors to carefully include the above discussions in a revised version of the paper, as also suggested by Reviewer 43yC.

---

> > ### Author Response · Authors · 2024-08-09
> >
> > Thank you very much for your feedback. We are grateful for the chance to improve the presentation of the current paper. For example, we will include a brief summary of the challenge and technical contribution after presenting Theorem 4.2. We will certainly carefully incorporate other relevant discussions into the revised version of the paper.
> >
> > We are especially grateful for your decision to increase the score, which is very encouraging for us.

---

> > ### Author Response · Authors · 2024-08-14
> >
> > Thank you for your valuable suggestions and for considering an increase in the score.  We appreciate your thoughtful review. (We noticed that the current score still appears as the previous one on our end, in case there might be an error.)
> >
> > We agree with your suggestions. After presenting Theorem 4.2, we will provide a brief introduction about the proof process of Theorem 4.2 and the previous results it builds upon. This will help clarify the challenges addressed by Theorem 4.2 and highlight our technical contributions . Also, we will include the above discussions in the revised version, and we would like to re-iterate the changes we will incorporate:
> >
> > 1. The proofs of Theorems 3.3 and 4.1 are standard, and therefore, we propose renaming them as propositions rather than theorems.
> >
> > 2. We will enhance the discussion of related work on mirror initialization in the introduction, ensuring that our findings are better integrated with existing research.
> >
> > 3. The revised version will emphasize the generalization advantages of mirror initialization over standard initialization, as indicated by our results.
> >
> > 4. We will correct minor errors, including spelling and grammar mistakes, and clarify any ambiguous expressions in the text.
> >
> > Thank you again for your thorough review and constructive feedback. If you have any further comments or questions, please feel free to ask. Thank you.

---

### Official Review · Reviewer_XKWd · 2024-07-12

**Soundness:** 3
**Presentation:** 2
**Contribution:** 3
**Rating:** 7
**Confidence:** 3

**Summary:**

The paper aims to study the impact of standard random weight initialization - as opposed to mirror initialization - on NTK theory.  The key observation is that, for standard initialization, the operation of the network at initialization $f_0$ acts as a bias on the regression function $f^*$.  When analyzing the convergence of the excess risk in the kernel gradient flow model the convergence rate depends on the smoothness of the regression function, which is (effectively) $f_0+f^*$.  The paper demonstrates that, regardless of how smooth $f^*$ is, the smoothness of $f_0+f^*$ is bounded above by the smoothness of $f_0$ (theorem 4.2).  Non-zero initialization thus places upper and lower bounds on the generalization error predicted by KGF (theorem 4.3 and 4.4).  This is claimed to bring into focus the shortcomings of KGF in the NTK context for analyzing the convergence of neural networks even in the infinitely wide limit.

**Strengths:**

It is obviously important to consider the role of standard random initialization on the training and convergence of neural networks.  The argument presented in the paper is clear and gives an interesting insight into the predictions of NTK and KGF for understanding generalization and convergence.  Mathematically the results are strong (as best as I can tell, having skimmed the supplementary).

**Weaknesses:**

Perhaps I am misreading the results, but it would seem to me that, far from demonstrating the downsides of NTK theory, you have in fact shown that NTK theory correctly predicts that standard initialization can degrade performance (in terms of the generalization error during training) compared to mirror initialization for the artificial data.  With regard the real data, all I can see is that you have shown that standard initialization will indeed decrease (effective) smoothness; however your predictions are asymptotic, so without a side-by-side comparison of standard and mirror initialization training (as per the artificial data) I don't see how this demonstrates anything at all.  It would certainly be interesting to see if the results for real data are the same as for artificial data as this would (presumably) indicate that mirror initialization is to be preferred where possible.

(fwiw this is the core reason for the discrepancy between my evaluation of soundness and my recommendation.  Mathematically the paper seems sound, but the interpretation seems incorrect to me: far from showing the downsides of NTK theory, the results would appear to show its power.  If I'm wrong or the paper is suitably modified then I am happy to upgrade my recommendation).

Minor point: please run this paper through a spell checker.

**Questions:**

My main question is whether or not it would be feasible to run mirror-initialization vs standard-initialization experiments on the real-world dataset, or even a high-dimensional artificial dataset.  My guess would be that the results would replicate the Figure 1, which could have interesting practical implications.

---

> ### Author Rebuttal · Authors · 2024-08-06
>
> Thanks a lot for your valuable suggestion. It greatly helps us re-interpret our main results. More precisely, Theorem 4.3 and 4.4 actually tell us that the generalization ability depends on the smoothness of the target function $f^*$ and the initialization function. In particular, given the target function $f^* \in [\mathcal H]^{s}, (s \geq \frac{3}{d+1})$,
>
> - The mirror initialization $f_0 = 0 \in [\mathcal H]^{\infty}$  and thus the generalization error is $n^{-\frac{s(d+1)}{s(d+1)+d}}$ ([li2023statistical]);
> - The standard initialization $f_0 \in [\mathcal H]^{\frac{3}{d+1}}$ and thus the generalization error is $n^{-\frac{3}{d+3}}$ (our result).
>
> Thus, the $L^2$ generalization error with mirror initialization is $n^{-\frac{s(d+1)}{s(d+1)+d}}$ (which is minimax optimal) and even if the target function $f^*$ is sufficiently smooth, the $L^2$ generalization error with standard initialization is $n^{-\frac{3}{d+3}}$ (which suffers the curse of dimensionality). i.e., the implicit bias introduced by initialization significantly impacts the generalization ability and we should prefer mirror initialization in practice.
>
> We reviewed previous literature and found that [zhang2020type] compared mirror initialization with standard initialization. In Section 7.2.3 and 7.2.4 of [zhang2020type], Figures 3 and 4 show experiments on the Boston house price dataset and the MNIST dataset, demonstrating that mirror initialization (i.e., referred to as Anti-Symmetric Initialization in his notation) indeed outperforms standard initialization in terms of generalization ability. This confirms that NTK theory correctly predicts the impact of initialization in real applications, and we will emphasize this point in our paper.
>
> In summary, we appreciate your highly valuable suggestion. We will revise our paper to emphasize the impact of different methods of initialization in the NTK theory. Thank you again for your insightful comments.
>
>
>
> ### References
>
>
>
> [li2023statistical] Li, Y., et al. (2024). “On the Eigenvalue Decay Rates of a Class of Neural-Network Related Kernel Functions Defined on General Domains”.
>
>
>
> [zhang2020type] Zhang, Y., et al. (2020). “A type of generalization error induced by initialization in deep neural networks”.

---

> > ### Comment · Reviewer_XKWd · 2024-08-13
> >
> > Thank you for this this clarification, I have raised my score accordingly.

---

> > > ### Author Response · Authors · 2024-08-13
> > >
> > > Thank you for your feedback and for taking the time to review our rebuttal. We are pleased to hear that our clarifications helped to address your concerns, and we greatly appreciate your willingness to increase the score. In response to your suggestions, we reconsider the interpretation of our results to ensure that the connection between NTK theory and the observed effects on generalization performance is well presented, and will highlight the potential of mirror initialization in the revised version.
> > >
> > > Thank you again for your thoughtful review and for helping us to improve the quality of our paper.

---

### Official Review · Reviewer_iQcQ · 2024-07-12

**Soundness:** 4
**Presentation:** 4
**Contribution:** 3
**Rating:** 6
**Confidence:** 4

**Summary:**

This paper explores the standard random mode of initialization of neural networks through the lens of the neural tangent kernel (NTK). This connection is made by showing that a randomly initialized neural network does indeed converge to the NTK uniformly during training thus allowing to analyze the generalization ability of a neural network with more relaxed assumptions on initialization. Furthermore, this paper finds that neural network performance under this regime comes at odds with the NTK theory which indicates that networks should perform poorly for high dimension inputs.

**Strengths:**

The paper is sufficiently original in that it explores the more conventional initialization schemes used with neural nets and shows that, despite uniform convergence to the NTK, the kernel itself contradicts real world performance of the networks. This is a continuation of the theme in NTK theory where the infinite width regime fails to fully capture the finite's performance.

This is presented in a clear and thorough manner with rigorous proofs and straightforward, but convincing experiments. In regards to significance, again, this further builds on top of work that has "poked holes" in the practical usage of NTK theory and as such is a relatively significant work in that direction.

**Weaknesses:**

There are a few typos/grammer mishaps that I caught in the paper:

Proposition 2.2 - "...noise term $\epsilon$ **satisfis**..."

Lines 126-127 - "...*let us reall the* concept..."

Line 233 - "This is **an** generalization..."

Lines 266-267 - "...the network **on longer generalizes** well..." (does not make sense)

Line 314 - "...datasets *from real world* and estimate the *smoothness of function*."

**Questions:**

Line 186 - Can you specify what space $C(\mathcal{X}, \mathbb{R})$ refers to?

Lines 193-194 - (Less of a question and more of a comment) I am assuming NNK is just your notation for the standard empirical NTK...

Lines 274-275 - Why is it necessary to remove the bias term for the derivation of the lower bound?

**Limitations:**

Everything is adequately addressed.

---

> ### Author Rebuttal · Authors · 2024-08-06
>
> ### Minors
>
> Thank you for your thorough review and for pointing out the typos and grammatical errors in our paper. We appreciate your feedback, and we have made the necessary corrections as follows:
>
> 1. **Proposition 2.2:** The typo "satisfis" has been corrected to "satisfies".
>     - Original: *"...noise term $\epsilon$ satisfis..."*
>     - Corrected: *"...noise term $\epsilon$ satisfies..."*
>
> 2. **Lines 126-127:** The typo "reall" has been corrected to "recall".
>     - Original: *"...let us reall the concept..."*
>     - Corrected: *"...let us recall the concept..."*
>
> 3. **Line 233:** The grammatical error "an generalization" has been corrected to "a generalization".
>     - Original: *"This is an generalization..."*
>     - Corrected: *"This is a generalization..."*
>
> 4. **Lines 266-267:** The phrase "on longer" has been corrected to "no longer" to clarify the meaning.
>     - Original: *"...the network on longer generalizes well..."*
>     - Corrected: *"...the network no longer generalizes well..."*
>
> 5. **Line 314:** The phrase has been corrected for clarity and grammatical correctness. The article "the" has been added before "real world" and "function" has been pluralized to "functions".
>     - Original: *"...datasets from real world and estimate the smoothness of function."*
>     - Corrected: *"...datasets from the real world and estimate the smoothness of functions."*
>
> We have carefully reviewed the entire manuscript to ensure that no additional errors are present. Thank you for your attention to detail, which has helped us improve the quality of our paper.
>
> ### Response to Questions
>
> 1. **Meaning of** $C(\mathcal{X},\mathbb{R})$:
> $C(\mathcal{X},\mathbb{R})$ includes all continuous functions from $\mathcal{X}$ to $\mathbb{R}$ with norm defined as $\lVert f \rVert = \sup_{x \in \mathcal{X}} |f(x)|$ for $f \in C(\mathcal{X},\mathbb{R})$. In fact, Lemma 3.2 on Line 186 is the Theorem 1.2 in [hanin2021random]. The definition of $C(\mathcal{X},\mathbb{R})$ is consistent with that of $C^0(T,\mathbb{R}^{n_L+1})$ in the latter. For more information on weak convergence of continuous random processes, we can refer to Section 7 of [billingsley2013convergence].
>
> 2. **NNK:** The Neural Network Kernel (NNK) we define here is indeed the standard empirical NTK defined in some other literature. Thank you for your feedback and we apologize for the confusion caused by our notation. We will specifically mention this point in the main text to avoid potential confusion caused by the name.
>
> 3. **Why Remove the Bias Term When Obtaining the Lower Bound:** This technical setting is for convenience in the derivation of our proof. After removing the bias term in network structure and further assuming the data is distributed on the sphere, the NTK is a dot-product kernel defined on sphere and the corresponding RKHS meets the so-called Regular RKHS condition (refer to Assumption 2 and Example 2.2 of [li2024generalization]). In general, this technical setting helps us obtain the lower bound of the generalization error.
>
> ### References
> [billingsley2013convergence] Billingsley, P. (2013). “Convergence of probability measures”.
>
> [hanin2021random] Hanin, B. (2021). “Random neural networks in the infinite width limit as Gaussian processes”.
>
> [li2024generalization] Li, Y., et al. (2024). “Generalization Error Curves for Analytic Spectral Algorithms under Power-law Decay”.

---

> > ### Comment · Reviewer_iQcQ · 2024-08-09
> >
> > Thank you for your comments and clarification. I concur with the other reviewers that the additional context you provided in your rebuttals would improve the quality of the paper. I will keep my score the same currently and await the revisions.
> >
> > I also found another typo on line 115 "... absolute positive constant $c$ **abd** $C$...".

---

> > > ### Author Response · Authors · 2024-08-10
> > >
> > > Thank you for your thoughtful feedback. I appreciate your agreement with the other reviewers regarding the additional context provided in the rebuttal and its potential to enhance the quality of the paper. I will carefully address the points raised and incorporate the necessary revisions.
> > >
> > > Additionally, thank you for pointing out the typo on line 115. I will correct the phrase to “absolute positive constant $c$ and $C$” in the revised version.
> > >
> > > I look forward to making these improvements and resubmitting the revised paper. Thank you for your careful review again.

---

### Official Review · Reviewer_43yC · 2024-07-13

**Soundness:** 3
**Presentation:** 2
**Contribution:** 3
**Rating:** 7
**Confidence:** 3

**Summary:**

This paper studies various kernel theories of neural networks, both random feature and neural tangent kernels. The main approach is to use the decay rates of the target function and kernel eigenvalues to get generalization error rates. The paper's novel theoretical contribution is to show uniform convergence of the network's input-output function $f$ to that of a kernel regressor with NTK kernel. The setting is that the weights are scaled to be in the NTK regime and when both are trained under gradient flow. Previously known results are used to characterize the RKHS associated with the NTK and "interpolations" of this space (which is useful if the target function isn't actually in the RKHS). It is argued that, since the network initialization with random weights is not smooth, that the network overall will have a slow error rate. These theoretical arguments are backed up with experiments on synthetic data comparing the standard initialization, that was studied theoretically, with a "mirrored" one. Also, the smoothness of the MNIST, CIFAR-10, and Fashion-MNIST classification problems is estimated and found to be quite different from the basic bound based on input dimensionality.

**Strengths:**

The work here brings together a bunch of connected kernel theory and neural networks results in a (relatively) readable treatement. The uniform convergence of the network's output to a Gaussian process is good to know. The overall goal of understanding the effect of initialization is certainly important, as is highlighting the weaknesses of kernel frameworks in understanding neural networks.

**Weaknesses:**

* In the introduction and experimental sections, the standard and mirror initializations are contrasted. From the introduction, I was expecting to see theoretical results that differentiated between the mirror and standard initializations. The authors could be clearer about what their contributions are and more clearly state what is known about the mirror initialization.
* As someone who is fairly familiar with kernel theory of neural networks, I still found it hard to identify which theorems and other results presented by the authors here are actually novel and which are applications of known results to their setting. For instance, isn't the decay rate of $1/(d+3)$ classical and known from Sobolev theory? I would suggest that the authors clarify this.
* The bulk of the paper is really just setting the stage for your results. I would have liked to see more explanation of how to interpret your results and discussion of how they fit into the greater understanding of these networks/kernels. I would also have liked more detail on the experiments within the main text.
* The synthetic experiments (Fig 1) only cover less than an order-of-magnitude in $n$. They would be much more convincing if they covered more. However, I can imagine you'd run into issues with double-descent and other factors that your asymptotic theory could have trouble with.

**Questions:**

* Please clarify the definition of the interpolation space (Eq 6 and around it). Your notation for the space of functions is unclear to me. Some explanation should be given. It is unusual to see a function written as a series $\sum_i a_i \lambda_i^{s/2} e_i(x)$
* Fix the formatting issue in line 104
* Prop 2.2 "satisfis" typo
* Do you think it would be more consistent to use the NTK superscript everywhere you treat it rather than f^NTK and K^NT ?
* Line 267 "network on longer" typo
* Does the mirror initialization lead to a significantly different error bound than yours, Thm 4.3?
* Can you include the method of estimating the smoothness of the various classification problems (Table 1) in the main text? Looking at the appendix, I see it depends on an empirical kernel/Gram matrix, but under what kernel is not clear. NTK?
* You report various least-square fits on log-log plots. You should be clear about how these were performed. Linear fits in log-space are known to have problems versus least-square fitting with power-laws. See the work by Clauset et al.
* In your notation section, you use standard $O, \Omega, o, \omega$ symbols but do not define the frequent $\asymp$ symbol. This should be explained.

**Limitations:**

The work here is very focused on the NTK theory of a multi-layer perceptron/fully-connected network. Strong claims are made about the weakness of such theories. However, different architectures lead to different kernels with different spectral properties and generalization performance. For instance, convolutional architectures lead to convolutional kernels which are known to work better in image classification and other spatially structured settings. It is also known that initialization with weights that are not Gaussian white noise can lead to better performance (see the work on "structured random features" and "rainbow networks").

I did not check the appendix proofs, but I read all of the main text results.

---

> ### Author Rebuttal · Authors · 2024-08-06
>
> ### Brief Introduction of Mirror Initialization
>
> *"In the introduction and experimental sections, the standard and mirror initializations are contrasted. ...... what is known about the mirror initialization."*
>
> Thank you for your suggestion. Here we provide a brief introduction of the existing theoretical results on mirror initialization.
>
> - Under mirror initialization, [lai2023generalization], [li2023statistical], and [lai2023generalization2] studied the generalization ability of single-layer fully connected, multi-layer fully connected, and multi-layer residual network, respectively, and proved the minimax optimality of network. For example, [li2023statistical], which studied the generalization ability of multi-layer fully connected mirror-initialized networks, concluded that for $f^* \in [\mathcal{H}^{NT}]^s, (s \geq 1)$, the $L^2$ generalization error is $n^{-\frac{s(d+1)}{s(d+1)+d}}$. This differs significantly from the $n^{-\frac{3}{d+3}}$ we obtained, reflecting the theoretical differences between different initialization.
>
> ### More Explanation of Our Contribution
>
>
> *"As someone who is fairly familiar with kernel theory of neural networks, I still found ...... I would suggest that the authors clarify this."*
>
>
> Before the response, we apologize for a typo. As mentioned in Theorem 4.2, the smoothness of the Gaussian process is $\frac{3}{d+1}$. Therefore, the smoothness in lines 250, 260, and 281 in the Theorem should also be $\frac{3}{d+1}$ instead of $\frac{1}{d+3}$. This typo does not affect our results.
>
> Thanks for your reminder and we are willing to respond to your questions. Since NTK and Gaussian process are well-studied objects, we do need to clarify what the novel contribution we have done here.
>
> The major contribution appears in Theorem 4.2 where we showed that $P(f^{GP} \in [\mathcal H^{NT}]^{t}) = 1_{t \textless \frac{3}{d+1}}$. Consequently, the generalization error of the gradient descent with standard random initialization is $n^{-\frac{3}{d+3}}$.
>
> To the best of our knowledge, this cannot be simply concluded from classical Sobolev theory.
>
> - **Challenge of Theorem 4.2:**
>   - If the input is uniformly distributed on the sphere $\mathbb{S}^d$ instead of $\mathcal{X}$, both NTK and RFK are inner-product kernels and their Mercer's decomposition shares the same eigen-function (i.e., spherical harmonic polynomials). In this situation, simple calculations on the eigenvalues would show that $f^{GP} \in [\mathcal{H}^{NT}]^{\frac{3}{d+1}}$.
>   - However, when the data is a general distribution supported in a bounded open set $\mathcal{X} \subset \mathbb{R}^d$, NTK and RFK will not possess the nice property anymore which makes the claim if $f^{GP} \in [\mathcal{H}^{NT}]^{\frac{3}{d+1}}$ unclear.
>   - Before we clarify our solution and technical contributions, we first recollect some notations. For a sub-region $S \subset \mathbb S^d$, let $\mathcal{H}_{0}^{NT}(\mathbb{S}^d)$ be the RKHS associated with the homogeneous NTK defined on $\mathbb S^d$ and $\mathcal{H}_0^{NT}(S)=\mathcal{H}_0^{NT}(\mathbb{S}^d) |_S$ be its restriction on the sub-region $S \subset \mathbb{S}^d$. We define $\mathcal{H}_0^{RF}(\mathbb{S}^d)$ and $\mathcal{H}_0^{RF}(S)$ similarly.
>   - We first show that, on a general $\mathcal{X}$, $\mathcal{H}^{RF} \cong [\mathcal{H}^{NT}]^\frac{d+3}{d+1}$ (where the $\cong$ is isomorphic as Hilbert space not RKHS).
>   - Let us fix a continuous and isomorphic mapping from $\mathcal{X} \subset \mathbb{R}^d$ to a subregion $S \subset \mathbb{S}^d$. We then show the equivalence (as RKHS) $$\mathcal{H}^{RF} \cong \mathcal{H}_{0}^{RF}(S), \mathcal{H}^{NT} \cong \mathcal{H}_0^{NT}(S).$$   This equivalence is not as trivial as it appears.
>   - We then need to show that $[\mathcal{H}_0^{RF}(S)] \cong [\mathcal{H}_0^{NT}(S)]^{\frac{d+3}{d+1}}$.
>   - Notice that the Mercer's decomposition of RFK and NTK share the same eigenfunctions (i.e., the spherical harmonic polynomials), we know that $[\mathcal{H}_0^{RF}(\mathbb S^d)] \cong [\mathcal{H}_0^{NT}(\mathbb{S}^d)]^{\frac{d+3}{d+1}}$. If the restriction $\mid_S$ and the interpolation $[]^{\frac{d+3}{d+1}}$ are commutative, then we are done. Unfortunately, this commutativity is not easy to verify in general.
>   - Fortunately, by a recent result [haas2023mind], we have $\mathcal{H}_0^{RF}(\mathbb{S}^d) \cong W^{\frac{d+3}{2},2}(\mathbb{S}^d)$ and $\mathcal{H}_0^{NT}(\mathbb{S}^d) \cong W^{\frac{d+1}{2},2}(\mathbb{S}^d)$ where $W^{s,2}$ denotes the usual fractional Sobolev space. Since Sobolev space is a more tractable object, we then carefully verify that $\mathcal{H}_0^{RF}(S) = [\mathcal{H}_0^{NT}(S)]^{\frac{d+3}{d+1}}$ and finish the proof. We would like to emphasize that there are several different definitions for the fractional Sobolev spaces and there are no similar results explicitly stated in the literature.
>   - By Lemma H.5, we have $[\mathcal{H}^{RF}]^{\frac{3}{d+3}} \cong [[\mathcal{H}^{NT}]^\frac{d+3}{d+1}]^{\frac{3}{d+3}} (\cong [\mathcal{H}^{NT}]^{\frac{3}{d+1}})$.
>   - If we denote $\mathcal{H}^{RF} = \left\lbrace \sum_{i \in \mathbb{N}} a_i \lambda_i^{\frac{1}{2}} e_i | \sum_{i \in \mathbb{N}} a_i^2 < \infty \right\rbrace$, then the K-L expansion gives us that $f^{GP} = \sum_{i \in \mathbb{N}} Z_i \lambda_i^{\frac{1}{2}} e_i$ where $\{Z_i\}$ are i.i.d. standard Gaussian variables. We can verify that $\mathbf{P}(f^{GP} \in [\mathcal{H}^{RF}]^{t}) = 1$ for $t < \frac{3}{d+3}$ and that $\mathbf{P}(f^{GP} \in [\mathcal{H}^{RF}]^{t}) = 0$ for $t \geq \frac{3}{d+3}$. i.e., we have
>   $\mathbf{P}(f^{GP} \in [\mathcal{H}^{NT}]^{t}) = 1$ for $t < \frac{3}{d+1}$ and that $\mathbf{P}(f^{GP} \in [\mathcal{H}^{NT}]^{t}) = 0$ for $t \geq \frac{3}{d+1}$.
>
> (We apologize for the inconvenience, due to the 6000-word limit, the second half is provided in the comment.)

---

> > ### Comment · Reviewer_iQcQ · 2024-08-09
> >
> > Hi, you mentioned that you included a second half to the rebuttal in another comment, but I don't think I can see it. It would be nice to read through it if possible/relevant.

---

> > > ### Author Response · Authors · 2024-08-10
> > >
> > > Hi, thank you for your message. I apologize for any confusion. I have now modified the comment to ensure that the second half of the rebuttal is visible to all reviewers. Please feel free to read through it, and I would greatly appreciate any further feedback you may have.

---

> ### Author Response · Authors · 2024-08-06
> **The second part of Rebuttal**
>
> (We apologize for reaching the word limit earlier, and we provide the second half of the rebuttal here.)
> ### Summary of the Theoretical Significance of the Paper
> *"The bulk of the paper is really just setting...... the experiments within the main text."*
>
> Thank you for your suggestion. We will summarize the theoretical significance of the paper within the background of NTK theory.
> - Background of our work: For a long time, NTK theory has played an important role in the study of network. Under NTK theory, a series of works tried to explain the remarkable generalization ability of network [lai2023generalization], [li2023statistical], [lai2023generalization2], which applied mirror initialization and derived the minimax optimality of network.
> - Impact of standard initialization: However, we wonder how the actual results would be if the standard initialization are considered. Under standard initialization, our result shows that the generalization ability of network under NTK theory is actually quite poor. It is a surprising result and prompts us to think further.
> - Significance of our work: On one hand, this tells us that the generalization ability of network is not only influenced by the smoothness of the goal function but also by the way of initialization. Thus, we have to carefully take the way of initialization into consideration. On the other hand, we can consider whether the inconsistency between the theory and reality is due to the overly strong assumptions of NTK theory (e.g., the width of network is sufficient large). In the future, it may be worthwhile to explore how NNK changes during training under finite width to advance the existing NTK theory.
> - Experiment: In the experiment of Section 5.1, we chose a single-layer network with width $m =20n$ to ensure the width is sufficiently large. At each $n$, we conducted 10 experiments with a learning rate of 0.6, and each training was run for 10n epochs to ensure sufficient time. We will add these details to the main text. Thank you for your suggestion.
> ### The Problem of Fig 1
> *"The synthetic experiments (Fig 1) only cover ..... theory could have trouble with."*
>
> Thanks for pointing this out and we appreciate your correction. Indeed, the range of our $n$ is not large enough. In kernel theory, it is common for the generalization error to decrease polynomially with respect to $n$ (e.g., Fig 1 in [li2024saturation]). In Fig 1, we aim to demonstrate a similar phenomenon. However, in fact, it indeed may encounter the problem of double-descent as you mentioned. To ensure the convergence of NTK, the width of the network $m$ needs to increase with $n$, but it will result in a high computational cost. The reasons above are why we didn't increase $n$. Thank you for correction again.
> ### Response to Questions
> 1. The notation $\sum a_i \lambda_i^{s/2}e_i(x)$ is mainly because $\lbrace\lambda_i^{s/2}e_i(x)\rbrace$ forms an ONB of the interpolation space. Thank you for pointing this out. We will explain this in more detail.
> 2. Under mirror initialization, for $f^*\in[\mathcal{H}^{NT}]^s$ ($s \geq 1$), the generalization error of network is $n^{-\frac{s(d+1)}{s(d+1)+d}}$, as shown in [li2023statistical]. We appreciate your reminder and we will add this point to the discussion of Theorem 4.3 for the convenience of comparison by readers.
> 3. Thank you for suggestion on the notations in the experimental part. We estimated the smoothness of real datasets with respect to the NTK of single-layer fully connected network. We will add this point in our paper.
> 4. Regarding the estimation of decay rate in the log-log plot, there may be some miscommunication. As a function of $n$, e.g., in Fig 1, we denote the experimentally measured generalization error as $e_i(n)$, where $n$ is the sample size and $i$ represents the $i$-th experiment. Then $e_i(n)$ is approximately of the form $e_i(n) \propto n^{-\alpha}Z_{n,i}$, where $Z_{n,i}$ denotes a random variable with values around 1, and $\alpha$ is the decay rate to be estimated. In this case, a linear fit after a log transformation may not be a bad choice. We found that Clauset et al. studied estimating $a$ through i.i.d. samples $x_{i} \sim p(x) (\propto x^{-a})$, which is slightly different from our problem. We guess there may be some miscommunication here.
> 5. Thanks for your careful reading again. We apologize for our annoying typos. We have tried our best to read through the paper and will make several modifications in the new revision.
> ### Impact of Network Architecture on Our Results
> Our results indeed focus on fully connected network. Thanks for pointing this out. For other architectures, like ResNet or CNN, if we know the properties of the corresponding NTK, RFK, and initial output function, it is also reasonable to apply our framework. This requires considerable preliminary work, but in the future, we are willing to explore whether our results hold under different network settings. Again, thank you for highlighting this point.

---

> ### Author Response · Authors · 2024-08-06
> **Refernces of Rebuttal**
>
> ### References
>
> [haas2023mind] Haas, M., et al. (2023). “Mind the spikes: Benign overfitting of kernels and neural networks in fixed dimension”.
>
> [lai2023generalization] Lai, J., et al. (2023). “Generalization ability of wide neural networks on \(\mathbb{R}\)”.
>
> [lai2023generalization2] Lai, J., et al. (2023). “Generalization ability of wide residual networks”.
>
> [li2024saturation] Li, Y., Zhang, H., and Lin, Q. (2024). “On the saturation effect of kernel ridge regression”.
>
> [li2023statistical] Li, Y., et al. (2024). “On the Eigenvalue Decay Rates of a Class of Neural-Network Related Kernel Functions Defined on General Domains”.

---

> ### Comment · Reviewer_43yC · 2024-08-09
>
> Thanks for your responses. I think it would be good if you used this time to clarify things in the final paper. I suggest you strengthen this context in the intro when you discuss related work and how your results fit in. I hope you plan to do this. I'll keep my score the same.

---

> > ### Author Response · Authors · 2024-08-09
> >
> > Thank you very much for your feedback and valuable suggestions on my rebuttal. I will take this opportunity to clarify the relevant issues in the final paper. Specifically, I will focus on strengthening the discussion of related work on mirror initialization in the introduction and better integrating my findings with existing research. I truly appreciate your guidance and will make the necessary improvements in my revisions. Thank you again for your review and support.

---

### Decision · Program_Chairs · 2024-09-25

**Decision:**

Accept (poster)

**Comment:**

This paper studies various kernel theories of neural networks, both random feature and neural tangent kernels. The presentation is clear, and contributions are incremental. However, the authors are still asked to polish this paper carefully based on responses.